# Everything in Its Place: Benchmarking Spatial Intelligence of Text-to-Image Models

**Zengbin Wang**[1,2*]**, Xuecai Hu**[1†]**, Yong Wang**[1†]**, Feng Xiong**[1]**, Man Zhang**[2]**, Xiangxiang Chu**[1]

[1]AMAP, Alibaba Group, [2]Beijing University of Posts and Telecommunications

⬡ Project Page: https://github.com/AMAP-ML/SpatialGenEval

## Abstract

Text-to-image (T2I) models have achieved remarkable success in generating high-fidelity images, but they often fail in handling complex spatial relationships, *e.g.*, spatial perception, reasoning, or interaction. These critical aspects are largely overlooked by current benchmarks due to their short or information-sparse prompt design. In this paper, we introduce **SpatialGenEval**, a new benchmark designed to systematically evaluate the spatial intelligence of T2I models, covering two key aspects: (1) SpatialGenEval involves 1,230 long, information-dense prompts across 25 real-world scenes. Each prompt integrates 10 spatial sub-domains and corresponding 10 multi-choice question-answer pairs, ranging from object position and layout to occlusion and causality. Our extensive evaluation of 23 state-of-the-art models reveals that higher-order spatial reasoning remains a primary bottleneck. (2) To demonstrate that the utility of our information-dense design goes beyond evaluation, we also construct another SpatialT2I dataset. It contains 15,400 text-image pairs with rewritten prompts to ensure image consistency while preserving information density. Fine-tuned results on current foundation models (*i.e.*, Stable Diffusion-XL, Uniworld-V1, OmniGen2) yield consistent performance gains (+4.2%, +5.7%, +4.4%) and more realistic effects in spatial relations, highlighting a data-centric paradigm to achieve spatial intelligence in T2I models.

## 1 Introduction

Recent developments in text-to-image (T2I) generation have demonstrated remarkable progress towards photorealistic and high-fidelity image generation (Zhang et al., 2023; Bie et al., 2024). This advancement has been largely driven by architectural innovations, evolving from early generative adversarial networks (GAN) (Goodfellow et al., 2020) to the dominant diffusion paradigms (Rombach et al., 2022). These paradigms are often augmented by powerful LLM text encoders (Black Forest Labs, 2024; Wu et al., 2025a) or integrated into unified multimodal architectures that merge generation and understanding capabilities (Deng et al., 2025; Chen et al., 2025b; OpenAI, 2024b; Lin et al., 2025; Xie et al., 2024; Chu et al., 2025b). Among these SOTA T2I models, a core success lies in their ability to render the fundamental '*what*' of a scene. They exhibit strong compositional capabilities in generating specified objects and binding them to their corresponding attributes (*e.g.*, color, material, shape), thus achieving high fidelity for basic semantic prompt following.

However, the limitations of these models become apparent when the task shifts from merely generating '*what*' is in a scene to precisely depicting '*where*' objects are located, '*how*' they are arranged, and '*why*' they interact within complex real-world scenes. As illustrated by the error cases in Figure 1, even SOTA T2I models (Wu et al., 2025a; Deng et al., 2025; OpenAI, 2024b) often fail on such fine-grained prompts. They may misplace objects, incorrectly orient them, disregard relative numerical comparisons, or fail to render causal interactions. These are not minor aesthetic flaws, but signal a fundamental shortcoming: *a lack of spatial intelligence* (Yang et al., 2025a)*, the core abilities of spatial perception, reasoning, and interaction with real-world scenes.*

Notably, these complex spatial failures are largely overlooked by current benchmarks. As in Figure 1 and Table 1, a significant number of current benchmarks (Wei et al., 2025; Ghosh et al., 2023; Huang

---

*Work done during the internship at AMAP, Alibaba Group.

†Project leads and corresponding authors.

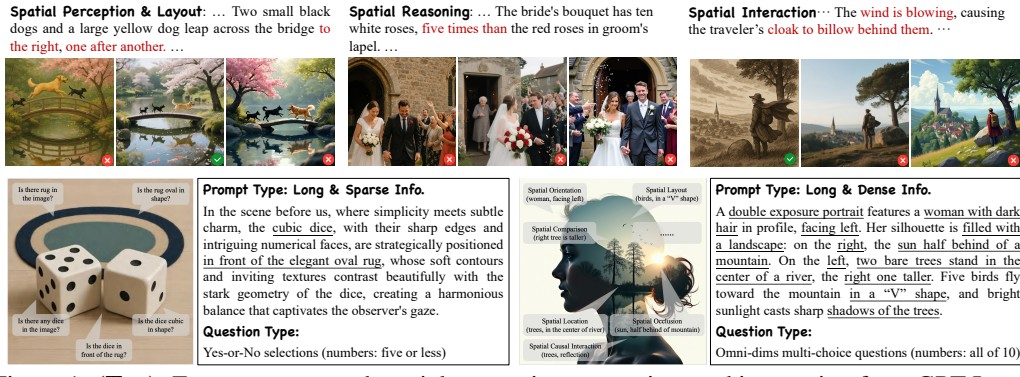

Figure 1: **(Top)**: Error cases around spatial perception, reasoning, and interaction from GPT-Image-1 (OpenAI, 2024b), Qwen-Image (Wu et al., 2025a), and Bagel (Deng et al., 2025). **(Bottom)**: A comparison of prompt and evaluation formats across current benchmarks (Wei et al., 2025).

et al., 2023) are structured around *short* or *information-sparse* prompts. This design inherently confines their scope to verifying the presence of objects, their attributes, or simple binary relations. Additionally, these are typically assessed using coarse-grained metrics, such as classification (Ghosh et al., 2023) or limited yes-or-no questions (Wei et al., 2025). Although valuable for assessing basic composition, these evaluations fall far short of probing a model's spatial capabilities, thus failing to capture critical deficiencies in higher-order reasoning or interaction. This highlights the need to explore long, information-dense, spatial-aware prompts and fine-grained evaluations for a comprehensive assessment of spatial intelligence.

To address this gap, we introduce **SpatialGenEval**, a new benchmark designed to systematically evaluate the spatial intelligence of T2I models. SpatialGenEval has two core features: (1) *Long & Information-dense & Spatial-aware Prompts*: SpatialGenEval involves a hierarchical decomposition of spatial intelligence into 4 domains (spatial foundation, perception, reasoning, and interaction), and 10 corresponding sub-domains, covering a comprehensive range of spatial abilities from object position and layout to occlusion and causality. Based on these, we construct 1,230 prompts across 25 real-world scenes. Each prompt is designed to integrate all 10 sub-domains, making it inherently long, information-dense, and suitable for comprehensive spatial evaluation. (2) *Omni-dimensional & Multi-choice Evaluations*: For fine-grained evaluations, each prompt is paired with 10 meticulously crafted omni-dimensional multiple-choice question-answer pairs, enabling precise identification of a model's successes and failures in all defined spatial capabilities.

To demonstrate our data's utility beyond evaluation, we follow the same principles of our SpatialGenEval to create another 1,230 prompts and corresponding 12,300 QAs. After filtering out low-quality design scenes from the initial 1,230 prompts, the remaining 1,100 prompts are sent to 14 top-performing open-source T2I models (accuracy > 50% on SpatialGenEval) for image generation. The resulting images are sent to the MLLM (*i.e.*, Qwen2.5-VL-72B) for evaluation, yielding a total of 15,400 text-image pairs. Subsequently, a powerful MLLM (*i.e.*, Gemini 2.5 Pro (Comanici et al., 2025)) refines their corresponding prompts to ensure better text-image alignment while preserving the original information density. Crucially, fine-tuning SOTA models like Stable Diffusion (Rombach et al., 2022), UniWorld-V1 (Lin et al., 2025), and OmniGen2 (Wu et al., 2025b) with this dataset significantly boosts their spatial intelligence, validating the data-centric approach as a viable and effective pathway for model improvement. In summary, our contributions are threefold:

- We introduce SpatialGenEval, a new benchmark to systematically evaluate complex spatial intelligence in T2I models. It leverages 1,230 information-dense prompts, each covering 10 spatial sub-domains and paired with 12,300 corresponding multiple-choice questions to evaluate a model's understanding beyond *what* to generate, to *where*, *how*, and *why*.

- Our extensive evaluation of 23 state-of-the-art models reveals a universal performance bottleneck in spatial reasoning. While models excel at basic object composition, their accuracy falls when faced with tasks requiring higher-order spatial understanding, such as relative positioning, occlusion, and causality, revealing this as a primary barrier to current T2I capabilities.

- Beyond evaluation, we explore a spatial-aware dataset (SpatialT2I), designed as a practical data-centric solution to improve the spatial intelligence of existing models. Fine-tuning results

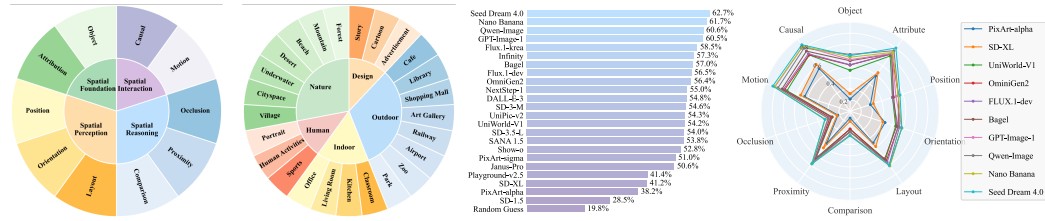

(a) 4 domains, 10 sub-domains  (b) 25 real-world scenes  (c) Overall Comparisons  (d) Detailed Comparisons

Figure 2: Overview of the SpatialGenEval benchmark and key results. The benchmark is structured around **(a)** 10 spatial sub-domains and **(b)** 25 real-world scenes. **(c)** The evaluation of 23 SOTA T2I models shows the overall performance ranking and **(d)** a detailed capability breakdown.

| Benchmarks | Prompt Length | Omni-Eval | QA Type | Object | Attribute | Position | Orientation | Layout | Comparison | Proximity | Occlusion | Motion | Causal |
|---|---|---|---|---|---|---|---|---|---|---|---|---|---|
| T2I-CompBench | S | ○ | Yes/No | ● | ● | ◑ | ○ | ○ | ○ | ○ | ○ | ● | ● |
| GenEval | S | ○ | Detect | ● | ● | ● | ○ | ○ | ● | ○ | ○ | ● | ○ |
| DPG-Bench | L | ○ | Score | ● | ● | ● | ○ | ○ | ○ | ○ | ○ | ○ | ○ |
| Wise | S | ○ | Score | ● | ● | ○ | ○ | ○ | ○ | ○ | ○ | ◑ | ◑ |
| TIIF-Bench | L | ○ | Yes/No | ● | ● | ● | ○ | ◑ | ◑ | ○ | ○ | ◑ | ○ |
| OneIG-Bench | L,S | ○ | Yes/No | ● | ● | ● | ○ | ◑ | ○ | ○ | ○ | ○ | ○ |
| SpatialGenEval | L | ● | Multi-Choice | ● | ● | ● | ● | ● | ● | ● | ● | ● | ● |

Table 1: Comparisons between our SpatialGenEval and current T2I Benchmarks. "L" and "S" denote long and short prompt. ●, ◑ , and ○ denote full, partial, and no coverage, respectively.

yield significant and consistent performance gains (+4.2% on Stable Diffusion-XL, +5.7% on UniWorld-V1, +4.4% on OmniGen2).

## 2 SPATIALGENEVAL BENCHMARK

In this section, we introduce SpatialGenEval, a new benchmark designed to evaluate the spatial intelligence of text-to-image models with dense information and omni-dimensional evaluations. An overview of SpatialGenEval is shown in Figure 2. Following, we first outline the key design principles of SpatialGenEval in Sec. 2.1. Subsequently, we present the focused spatial aspects and their definitions in Sec. 2.2, ranging from object position and layout to occlusion and causality. Finally, we detail the full benchmark construction pipeline in Sec 2.3, covering both the generation of information-dense prompts and their corresponding 10 omni-dimentional question-answer pairs.

### 2.1 KEY PRINCIPLES

- **Long & information-dense prompts**: A primary limitation of current T2I benchmarks is their reliance on *short* or *information-sparse* prompts (Wei et al., 2025; Ghosh et al., 2023; Huang et al., 2023; Niu et al., 2025), often confined to simple object-attribute pairs or simple relations. To better capture the complexity of real-world scenes and probe a model's ability to synthesize intricate information, SpatialGenEval is designed to utilize *longer* and *information-dense* prompts that are densely packed with multiple and interdependent spatial constraints.
- **Omni-dimensional multi-choice questions**: Instead of coarse-grained metrics about objects, attributions, and simple spatial relations in a yes-or-no selection (*e.g.*, "Is there any dice in the image?"), we evaluate models across all distinct sub-domains of spatial intelligence in a multi-choice format. For each prompt, 10 multi-choice questions across all dimensions are generated, allowing for a fine-grained diagnosis of where a model succeeds or fails.
- **Image-dependent answer (no answer leakage)**: Recent MMStar (Chen et al., 2024c) has revealed a significant flaw in MLLMs: some MLLMs can generate answers without accessing images. For this case, we do not send the "text-to-image prompt" to the evaluator to prevent "answer leakage". Additionally, the questions are also checked with humans to avoid "answer leakage".
- **Refuse to answer (not guess):** For each multiple-choice question, the MLLM evaluation task is instructed to select the best-matching option. To avoid cases where the model is forced to select an incorrect option when all options (A/B/C/D) are not faithful to the generated image (Yu et al., 2023), we include another "E: None" option to account for such generation failures.

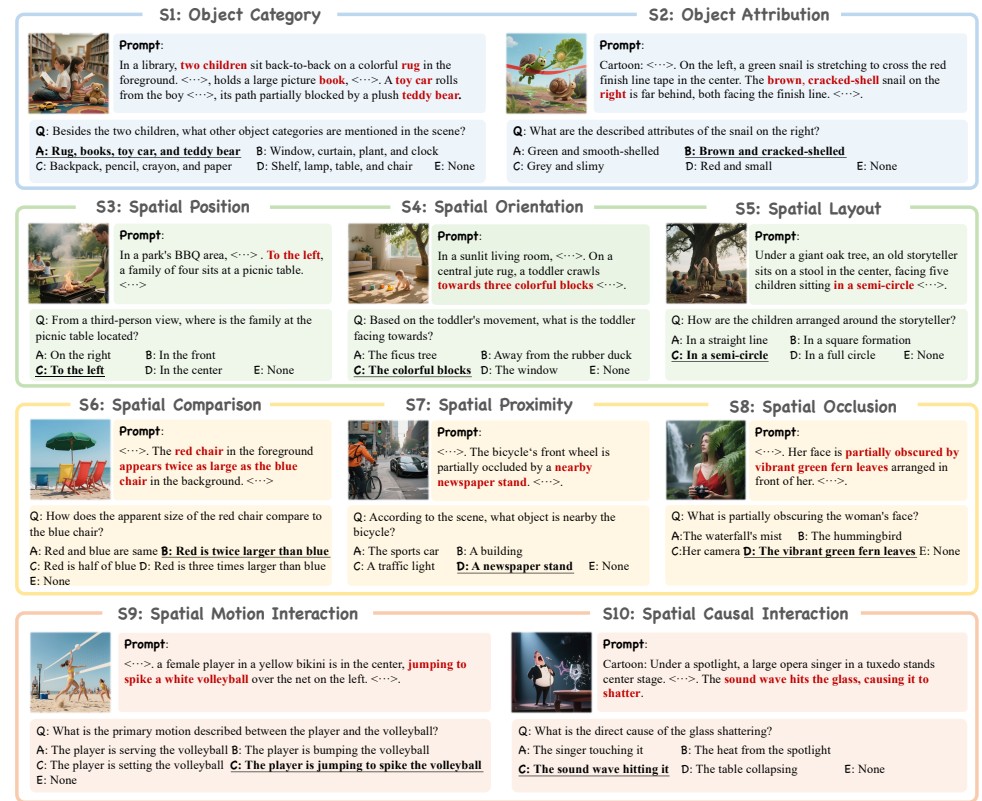

Figure 3: Examples of SpatialGenEval. Each image is generated from an information-dense prompt covering all 10 spatial sub-domains and evaluated with 10 corresponding multiple-choice questions.

## 2.2 FOCUSED ASPECTS

Motivated by advances in spatial cognition (Ruan et al., 2025; Malanchini et al., 2020) and recent studies about spatial intelligence (Yang et al., 2025b;a; Cai et al., 2025; Stogiannidis et al., 2025), the definition of spatial intelligence has evolved to include perception, reasoning, and interaction with the environment. Following this comprehensive view, we define spatial intelligence in Spatial-GenEval through a hierarchical framework that begins with Spatial Foundation (representing objects and their attributions), moves to Spatial Perception (perceiving their arrangement in space), advances to Spatial Reasoning (inferring relationships between them), and culminates in Spatial Interaction (understanding dynamic events and their causes). Representative examples are shown in Figure 3.

**Spatial Foundation (S1/S2).** This domain evaluates the model's ability to generate semantically correct objects, focusing on compositional completeness and attribute binding, including: **(S1) Object Category** evaluates compositional completeness by testing the model's ability to generate mentioned objects without omission or hallucination. **(S2) Object Attribution** evaluates attribute binding by examining whether the model correctly assigns attributes (*e.g.*, color, shape, material) to their designated objects, preventing attribute leakage.

**Spatial Perception (S3/S4/S5).** Building upon the foundational generation of objects, this domain evaluates the model's ability to interpret and render its geometric and relational arrangements on the 2D canvas. It focuses on the accurate translation of spatial language into visual form, through three sub-dimensions: **(S3) Spatial Position** evaluates the localization of an object using absolute (*e.g.*, top-left, bottom) or relative (*e.g.*, to the right of the book, to his left side) terms. **(S4) Spatial Orientation** focuses on rotational alignment, such as generating objects with specified facing directions (*e.g.*, facing left, upside down). A common challenge is that models often lean towards default poses (*e.g.*, front view). **(S5) Spatial Layout** assesses the model's understanding of multi-object arrangements. This extends beyond individual positions to collective configurations, such as linear sequences (*e.g.*, in a line from left to right), circular formations, or other specified group structures. This sub-dimension is crucial for testing the comprehension of group-level spatial patterns.

**Spatial Reasoning (S6/S7/S8).** This domain moves beyond direct perception to assess a model's higher-order cognitive ability to understand and render abstract, implicit, and 3D-aware spatial rela-

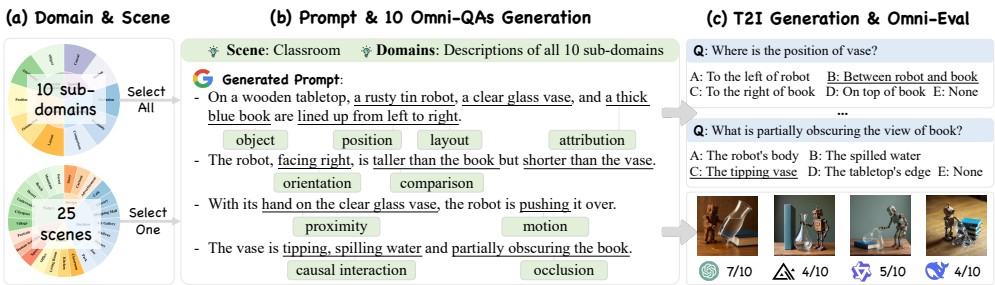

Figure 4: SpatialGenEval Construction Pipeline. **(a)** The process begins by selecting one of 25 real-world scenes and combining it with the definitions of all 10 spatial sub-domains. **(b)** The MLLM sequentially synthesizes an information-dense prompt that integrates all 10 constraints, along with 10 corresponding omni-dimensional QA pairs. **(c)** T2I models generate an image from the prompt, which is then evaluated against the QA pairs to yield a fine-grained spatial intelligence score.

tionships, involving: **(S6) Spatial Comparison** evaluates the model's grasp of relative quantitative attributes. This involves generating objects that adhere to comparative statements about their properties, such as size (*e.g.*, three times taller than), quantity, or length. This tests whether the model can perform reasoning rather than merely generating objects in isolation. **(S7) Spatial Proximity** focuses on the fine-grained physical distance between objects. It challenges the model to render precise interactions like "touching", "closest to", or "far from". This subdimension is critical for assessing the model's ability to control object boundaries and depict intimate spatial relationships, which are often overlooked in favor of simple co-occurrence. **(S8) Spatial Occlusion** assesses the model's implicit understanding of 3D scene structure and depth. This requires generating a scene where one object partially or fully obscures another (*e.g.*, the vase is partially obscuring the book). Success in this area indicates a more sophisticated world model that can reason about viewpoint and object layering, moving beyond a flat, 2D composition.

**Spatial Interaction (S9/S10).** This is the most advanced domain, evaluating the model's ability to depict dynamic events and physical causality. It moves beyond static scene composition to test whether a model possesses a rudimentary understanding of physics and temporal progression. This capability is divided into two distinct but related forms of interaction: **(S9) Spatial Motion Interaction** focuses on generating objects in dynamic states or mid-action sequences. It requires capturing a specific temporal moment, such as "a dog jumping over a log" or "a mid-flight ball". This tests the model's ability to convey movement through pose, trajectory, and contextual cues rather than relying on static or canonical placements. **(S10) Spatial Causal Interaction** evaluates the capacity of the model to illustrate explicit cause-effect relationships between objects or environments. Examples include "a rock hitting water and causing ripples" or "a hammer striking a nail into wood". Success in this dimension implies that the model can reason about functional physical relationships and translate latent dynamics into visually consistent and logically plausible images.

## 2.3 BENCHMARK CONSTRUCTION

Following the definition of 10 spatial-aware aspects, the construction for the SpatialGenEval benchmark involves two main stages: information-dense & spatial-aware prompt generation (Sec. 2.3.1) and the generation of their corresponding omni-dimensional question-answer pairs (Sec. 2.3.2).

### 2.3.1 INFORMATION-DENSE & SPATIAL-AWARE PROMPT GENERATION

**Automated prompt generation.** As illustrated in Figure 4, we instruct Gemini 2.5 Pro (Comanici et al., 2025) using two inputs: a specific scene from a curated set of 25 real-world scenes (*i.e.*, nature, indoor, outdoor, human, and design, as detailed in Appendix A.1) and the definitions of 10 spatial sub-domains. The model's task is to seamlessly integrate all 10 spatial constraints into a single, fluent, and logically sound prompt based on one given scene. The target length is set to approximately 60 words, balancing compatibility (77 tokens) with CLIP encoders (Radford et al., 2021) and the need for high information density. The meta instruction is shown in the Appendix A.4.

**Human-in-the-loop refinement.** While powerful, MLLMs can generate prompts that are stylistically awkward, logically unsound, or unfairly complex. To guarantee prompt quality, each MLLM-generated output is meticulously reviewed by human experts with the same guidebook. For example,

(1) disjointed phrases like "There is a robot. It is rusty." will be combined into a natural phrase like "A rusty robot". (2) Logical impossibilities like a cyclical layout ("A is left of B, B is left of C, C is left of A") will be identified and corrected. (3) Furthermore, to ensure that our focus is on spatial reasoning rather than lexical knowledge, ambiguous or unusual words (*e.g.*, vermilion, nostalgic, futuristic, bustling, tranquil) will be removed or replaced with more common synonyms (*e.g.*, vermilion → bright red). Totally, this validation process yields our final set of 1,230 high-quality, information-dense prompts across 25 real-world scenes. More details are shown in Appendix A.2.

### 2.3.2 OMNI-DIMENSIONAL QAS GENERATION

**Automated omni-dimensional QAs generation.** For each of the 1,230 prompts, we instruct a powerful Multimodal Large Language Model (*i.e.*, Gemini 2.5 Pro) to automatically generate 10 multiple-choice questions, with each question targeting exactly one of the 10 spatial sub-domains. To guide the model effectively, its input for each prompt includes three parts: the prompt, the definitions of all 10 spatial sub-domains, and a set of example questions. The model is then instructed to generate all 10 QA pairs at once. Each generated pair consists of the question, a ground-truth answer drawn from the prompt, and three other plausible but incorrect options designed to test a model's detailed understanding. The meta instruction is shown in the Appendix A.4.

**Human-in-the-loop refinement.** Following automated generation, every QA pair undergoes a rigorous human validation process with the same guidebook to confirm that: (1) To prevent answer leakage, the explicit task of human annotators is to identify and eliminate the question containing an explicit answer, where the answer lies in the question. For example, the question "What is the layout of the leaves that are arranged in a circle?" should be revised into "What is the layout of the leaves in the image?". (2) Refuse to answer (not guess): After the above validation, we programmatically append a "E: None" option in each question. This allows the evaluator to refuse a forced choice when none of the options are faithful to the generated image. More details are shown in Appendix A.2.

## 3 EXPERIMENTAL RESULTS

### 3.1 SETUP

**Text-to-Image models.** We test a wide range of well-known text-to-image models with diverse model architectures and model scales, covering 23 open-source and closed-source models: (1) **Diffusion Models**: Stable-Diffusion-series (Rombach et al., 2022), PixArt-series (Chen et al., 2023; 2024b), Flux-series (Black Forest Labs, 2024), playground-v2.5 (Li et al., 2024), SANA-1.5 (Xie et al., 2025), and more recent Qwen-Image (Wu et al., 2025a). (2) **AutoRegressive Models**: OmniGen2 (Wu et al., 2025b), NextStep-1 (NextStep Team et al., 2025), and Infinity (Han et al., 2025). (3) **Unified Models**: Janus-Pro (Chen et al., 2025b), Show-o (Xie et al., 2024), UniWorld-V1 (Lin et al., 2025), UniPic-v2 (Skywork Multimodality Team, 2025), and Bagel (Deng et al., 2025). (4) **Closed-source Models**: DALL-E-3 (Ramesh et al., 2021), GPT-Image-1 (OpenAI, 2024b), Nano Banana (Gemini-2.5-Flash-Image) (Google, 2025), and Seed Dream 4.0 (ByteDance, 2025).

**MLLM as a judge.** If not specific, we formulate the evaluation as a zero-shot, multiple-choice VQA task, using an open-source MLLM (*i.e.*, Qwen2.5-VL-72B (Bai et al., 2025)) as the primary evaluator. This choice leverages the model's SOTA capabilities while ensuring long-term reproducibility and avoiding reliance on closed-source APIs. For a comparison with closed-source MLLM, we also evaluate with GPT-4o-250306 (OpenAI, 2024a), and the results are presented in Appendix A.3.

**Evaluation details.** (1) For the evaluation, the MLLM evaluator is presented with a generated image and its 10 corresponding questions from 10 sub-domains. The evaluator's task is to select the most accurate description from five options: four plausible choices (A-D) derived from the prompt and a crucial fifth option, "E: None". To enhance the stability of this automated evaluation and reduce randomness, we implement a 5-round voting mechanism (Wang et al., 2022). A response is considered correct only if the MLLM selects the ground-truth answer in at least *4 of the 5 rounds*. The final score is then reported as the accuracy on each of the 10 spatial sub-domains, calculated as the percentage of correctly answered questions. (2) For time cost, we conduct the evaluation on $8 \times$H20 GPUs and deploy vllm framework (Kwon et al., 2023) in local environment. The evaluation time cost is around 1.8 seconds per image, completing all 1230 generated images in about 40 minutes.

| Model | Size | Overall | Spatial Foundation | | Spatial Perception | | | Spatial Reasoning | | | Spatial Interaction | |
|---|---|---|---|---|---|---|---|---|---|---|---|---|
| | | | Object | Attribute | Position | Orientation | Layout | Comparison | Proximity | Occlusion | Motion | Causal |
| Random | - | 19.8 | 20.1 | 19.3 | 19.8 | 19.8 | 19.7 | 20.3 | 19.5 | 19.6 | 20.1 | 19.8 |
| *1. Diffusion Generative Model* | | | | | | | | | | | | |
| SD-1.5 | 0.86B | 28.5 | 8.5 | 33.7 | 19.5 | 29.2 | 38.2 | 12.8 | 37.7 | 15.6 | 42.0 | 47.6 |
| PixArt-alpha | 0.6B | 38.2 | 20.9 | 49.3 | 29.2 | 43.1 | 45.3 | 16.2 | 45.9 | 21.5 | 52.9 | 57.3 |
| SD-XL | 3.5B | 41.2 | 25.7 | 52.8 | 32.0 | 40.9 | 49.3 | 19.1 | 50.7 | 22.4 | 56.7 | 62.0 |
| Playground-v2.5 | 2.5B | 41.4 | 27.0 | 55.8 | 31.9 | 41.5 | 49.0 | 18.5 | 49.8 | 22.4 | 55.2 | 63.3 |
| PixArt-sigma | 0.6B | 51.0 | 42.8 | 67.6 | 43.7 | 49.1 | 58.0 | 25.9 | 57.6 | 27.0 | 67.0 | 71.0 |
| SD-3-M | 2B | 54.6 | 52.6 | 72.1 | 46.9 | 50.7 | 62.4 | 26.1 | 62.7 | 27.4 | 71.4 | 74.1 |
| SD-3.5-L | 8B | 54.0 | 52.4 | 72.0 | 44.7 | 52.0 | 62.7 | 25.4 | 61.3 | 27.4 | 69.4 | 72.6 |
| SANA 1.5 | 4.8B | 53.8 | 48.5 | 70.0 | 47.3 | 51.1 | 62.2 | 25.9 | 59.9 | 28.5 | 70.0 | 75.0 |
| FLUX.1-dev | 12B | 56.5 | 51.7 | 73.8 | 50.0 | 55.5 | 66.7 | 28.2 | 62.9 | 28.9 | 73.1 | 73.8 |
| FLUX.1-krea | 12B | 58.5 | 58.0 | 75.4 | 50.7 | 55.7 | 67.1 | 28.3 | 66.7 | 28.0 | 76.0 | 78.8 |
| Qwen-Image | 20B | 60.6 | 61.0 | 77.2 | 55.6 | 56.7 | 69.7 | 28.6 | 67.7 | 30.8 | 78.1 | 80.2 |
| *2. AutoRegressive Generative Model* | | | | | | | | | | | | |
| NextStep-1 | 14B | 55.0 | 45.4 | 69.0 | 46.7 | 52.3 | 64.0 | 26.7 | 62.5 | 32.0 | 73.7 | 77.4 |
| OmniGen2 | 4B | 56.4 | 51.5 | 73.6 | 55.9 | 55.5 | 65.4 | 26.0 | 64.2 | 27.3 | 72.0 | 72.6 |
| Infinity | 8B | 57.4 | 53.7 | 73.0 | 53.7 | 57.5 | 65.2 | 27.9 | 64.5 | 29.1 | 72.6 | 76.3 |
| *3. Unified Generative Model* | | | | | | | | | | | | |
| Janus-Pro | 7B | 50.6 | 30.9 | 62.0 | 43.2 | 47.4 | 60.2 | 26.3 | 60.2 | 31.5 | 70.2 | 74.3 |
| Show-o | 1.3B | 52.8 | 41.7 | 68.3 | 46.7 | 48.2 | 60.8 | 26.6 | 61.1 | 28.9 | 69.5 | 75.8 |
| UniWorld-V1 | 12B | 54.2 | 46.8 | 71.3 | 50.1 | 53.1 | 64.0 | 26.1 | 62.0 | 26.8 | 69.6 | 72.4 |
| UniPic-v2 | 9B | 54.3 | 41.4 | 69.1 | 44.9 | 51.0 | 63.3 | 27.8 | 63.1 | 30.0 | 75.1 | 77.1 |
| Bagel | 7B | 57.0 | 55.3 | 73.7 | 51.2 | 54.0 | 62.9 | 28.6 | 64.1 | 29.0 | 74.4 | 76.7 |
| *4. Closed-source Generative Model* | | | | | | | | | | | | |
| DALL-E-3 | - | 54.8 | 51.1 | 67.9 | 41.5 | 52.9 | 63.3 | 28.4 | 62.4 | 28.0 | 75.2 | 77.4 |
| GPT-Image-1 | - | 60.5 | 56.3 | 74.1 | 53.3 | 58.9 | 70.4 | 31.4 | 66.8 | 30.2 | 80.9 | 82.2 |
| Nano Banana | - | 61.7 | 58.5 | 75.3 | 55.5 | 58.9 | 70.9 | 31.8 | 68.7 | 33.5 | 81.4 | 82.2 |
| Seed Dream 4.0 | - | 62.7 | 59.9 | 80.2 | 57.2 | 58.9 | 70.1 | 32.1 | 68.3 | 33.8 | 83.0 | 83.8 |

Table 2: SpatialGenEval leaderboard based on Qwen 2.5 VL (72B). "Random": random selection.

## 3.2 MAIN RESULTS OF SPATIALGENEVAL BENCHMARK

**Overall findings of SpatialGenEval.** Building on the overall leaderboard results (Table 2) of SpatialGenEval across 23 open-source and closed-source generative models, several key findings are revealed regarding model performance, core weaknesses, and promising development strategies.

- **Open-source models are catching up to closed-source ones, yet spatial intelligence remains a significant challenge.** Across all models, the overall performance on SpatialGenEval generally reflects the continuous improvement of SOTA T2I models over time. Notably, the gap between open-source and closed-source models is narrowing, with the best open-source model, Qwen-Image (60.6%), now catching up to the leading closed-source model, Seed Dream 4.0 (62.7%). Despite this progress, the highest score remains around the 60-point passing threshold, highlighting that even SOTA models possess only a rudimentary grasp of complex spatial intelligence.

- **Imbalanced performance between spatial foundation and higher-order spatial intelligence.** A key finding across all models is the performance gap between basic and advanced spatial skills. For spatial foundation tasks, top models like Qwen-Image and Bagel score above 70.0% on object and attribute generation. However, the performance drops on tasks that require complex thinking in complex spatial perception, reasoning, and interaction. This suggests that while models can draw objects correctly, they struggle to organize them according to specific rules.

- **Spatial reasoning emerges as the primary bottleneck.** Notably, spatial reasoning is the main weakness across all spatial domains. Scores for subtasks like comparison and occlusion are often below 30%, near the random selection (20%). This reveals a core failure of current T2I models: they can render objects correctly but cannot bind the semantic properties of objects to the structural logic of a scene, such as relative size or physical layering. Moreover, the most gap between open-source and closed-source models also lies in the spatial reasoning and interaction sub-domains.

- **Text encoder capability emerges as a key determinant of spatial intelligence.** Our results reveal a clear trend where models with stronger text encoders, particularly those leveraging powerful LLMs, consistently outperform those with standard CLIP encoders. For example, the top-performing open-source model, Qwen-Image (60.6%), utilizes a powerful LLM encoder. Similarly, models that enhance the standard CLIP architecture with more advanced encoders, such as FLUX.1 (56.5-58.5%) and SD-3 (54.0-54.6%) using T5, significantly outperform older models reliant solely on CLIP, like SD-1.5 (28.5%). This strongly suggests that a deeper understanding of complex, information-dense prompts is critical to achieve high-fidelity spatial generation.

Figure 5: Distribution of error types across scenes (left, based on all T2I models) and some examples of T2I models (right) in our SpatialGenEval.

| Model | GenEval | DPG-Bench | Wise | TIIF-Bench | Meta-Rank | SpatialGenEval (Ours) | Rank (Ours) |
|---|---|---|---|---|---|---|---|
| Janus-Pro | 0.80 | 84.19 | 0.35 | 65.02 | 5 | 50.60 | 5 |
| SD-3.5-L | 0.71 | 84.08 | 0.46 | 66.96 | 4 | 54.00 | 4 |
| Flux.1-dev | 0.82 | 83.84 | 0.50 | 71.78 | 2 | 56.50 | 3 |
| Bagel | 0.82 | 67.20 | 0.52 | 71.70 | 3 | 57.00 | 2 |
| Qwen-Image | 0.91 | 88.32 | 0.62 | 86.83 | 1 | 60.60 | 1 |

Table 3: Meta-ranking consistency of five popular models on two commonly used text-to-image benchmarks (GenEval (Ghosh et al., 2023), DPG-Bench (Hu et al., 2024)), two newly proposed benchmarks (Wise (Niu et al., 2025), TIIF-Bench (Wei et al., 2025)), and our SpatialGenEval.

- **Model scale and architecture are two possible pathways for advanced spatial intelligence.** Our results also reveal two concurrent trends driving improvements in spatial intelligence. (1) For model scale, specialized diffusion models generally follow a trend where performance correlates with model size. For example, the 20B Qwen-Image (60.6%) significantly outperforms the 8B SD-3.5-L (54.0%). (2) For model architecture, unified models demonstrate greater parameter efficiency by integrating understanding and generative abilities. For example, the 7B Bagel model (57.0%) achieves a score comparable to the much larger 12B FLUX.1-krea (58.5%). This highlights their potential to advance spatial intelligence without relying solely on parameter count.

**Failure cases analysis.** Our analysis in Figure 5 provides a detailed breakdown of failure cases across scenes and models. Models first face Basic Composition and generally succeed with low error rates. The next challenge of Visual Perception is more difficult. Its errors are highest in complex Nature scenes at 28.5%. A significant increase in difficulty then occurs with Relational Reasoning. This skill is the primary failure point for all models, and its error rates often exceed 35%. Interestingly, Motion Interaction is a less frequent source of error, with rates typically remaining below 18%. This distribution suggests that the principal barrier to achieving advanced spatial intelligence is not a linear progression of skills, but a critical weakness in processing relational logic.

**Correlation with other benchmarks.** In Table 3, the model rankings on SpatialGenEval closely align with the meta-rankings from four other major benchmarks. This strong correlation validates our benchmark as a reliable indicator of a model's overall generative capability.

**Other evaluation models as the judge.** To test for judge-dependency, we evaluate models using both GPT-4o (OpenAI, 2024a) and Qwen2.5-VL-72B (Bai et al., 2025). As shown in Table 4, both judges produce similar model rankings and numbers. The consistency of the similar ranking and number validates the robustness of our benchmark and the selected evaluator, demonstrating that its relative results are not biased by the choice of evaluator.

| Model | GPT-4o-250306 | | Qwen2.5-VL (72B) | |
|---|---|---|---|---|
| | Overall | Rank | Overall | Rank |
| Janus-Pro | 48.0 | 5 | 50.6 | 5 |
| SD-3.5-L | 52.9 | 4 | 54.0 | 4 |
| Flux.1-dev | 54.3 | 3 | 56.5 | 3 |
| Bagel | 56.6 | 2 | 57.0 | 2 |
| Qwen-Image | 60.8 | 1 | 60.6 | 1 |

Table 4: Overall ranking consistency of closed- and open-source evaluations on SpatialGenEval.

**Human Alignment Study.** To evaluate the effectiveness of MLLMs as evaluators, we conduct a human alignment study with three top-performing models: the open-source Qwen2.5-VL-72B and the closed-source GPT-4o and Gemini-2.5-Pro. We randomly sample 200 images from Qwen-Image (8 from each of 25 scenes). Following the principles in Section 2.1, five human annotators work independently to select

| MLLMs | Foundation | Perception | Reasoning | Interaction | Overall |
|---|---|---|---|---|---|
| Qwen2.5-VL-72B | 87.0 | 76.5 | 73.3 | 84.8 | 80.4 |
| GPT-4o | 82.8 | 76.7 | 72.5 | 83.3 | 78.8 |
| Gemini-2.5-Pro | 91.0 | 81.5 | 78.2 | 86.0 | 84.2 |

Table 5: Human alignment study across open-source and closed-source MLLMs based on the balanced accuracy (%).

| Model | Overall | Spatial Foundation | | Spatial Perception | | | Spatial Reasoning | | | Spatial Interaction | |
|---|---|---|---|---|---|---|---|---|---|---|---|
| | | Object | Attribute | Position | Orientation | Layout | Comparison | Proximity | Occlusion | Motion | Causal |
| SD-XL | 41.2 | 25.7 | 52.8 | 32.0 | 40.9 | 49.3 | 19.1 | 50.7 | 22.4 | 56.7 | 62.0 |
| + SptialT2I | 45.4 | 29.0 | 55.7 | 38.6 | 48.3 | 53.6 | 23.0 | 54.8 | 27.2 | 61.6 | 63.2 |
| UniWorld-V1 | 54.2 | 46.8 | 71.3 | 50.1 | 53.1 | 64.0 | 26.1 | 62.0 | 26.8 | 69.6 | 72.4 |
| + SptialT2I | 59.9 | 50.8 | 74.8 | 58.5 | 63.5 | 70.4 | 34.6 | 70.7 | 31.6 | 66.0 | 78.5 |
| OmniGen2 | 56.4 | 51.5 | 73.6 | 55.9 | 55.5 | 65.4 | 26.0 | 64.2 | 27.3 | 72.0 | 72.6 |
| + SptialT2I | 60.8 | 61.8 | 71.8 | 62.1 | 59.9 | 67.2 | 35.3 | 66.3 | 34.2 | 74.5 | 74.8 |

Table 6: Quantitative fine-tuned results of recent T2I models on SpatialGenEval.

**Prompt 1**: In a forest, a grey squirrel on a left branch of a massive oak tree drops an acorn towards a red squirrel at the bottom-center. The two squirrels face each other. Above them, a wise-looking owl with yellow eyes, partially obscured by its tree hole, faces the observer. The oak tree is much taller than a nearby mossy log. The red squirrel is close to a small pile of three brown acorns, illuminated by a sunbeam.

GPT-Image-1 · Qwen-Image · Flux.1 dev · Bagel · UniWorld-V1 · OmniGen2 · +SpatialT2I

**Prompt 2**: In a forest clearing, a family of three—two adults with a child between them—sits on a red and white checkered blanket. They face away from the viewer, looking up at fireworks. A golden retriever lies next to the child, and a picnic basket is on the blanket's right side. A tall tree on the left partially blocks the fireworks. One adult pulls the child close as a gentle wind rustles the leaves.

Figure 6: Qualitative comparisons of recent T2I models on SpatialGenEval.

the best option based solely on the given image and 10 questions, with no access to the original text-to-image prompt to prevent "leakage". We measure alignment using balanced accuracy (Brodersen et al., 2010), following (Li et al., 2025). Table 5 shows that all MLLMs align well with human judgment, with Gemini-2.5-Pro performing best. Moreover, alignment correlates with sub-domain difficulty of each sub-domain. The alignment is higher on simpler dimensions like Spatial Foundation/Perception/Interaction, while lower on the Spatial Reasoning sub-domain. Despite this, the alignment score still nears 80%, validating their effectiveness as evaluators in SpatialGenEval.

## 4 SUPERVISED FINE-TUNING (SFT)

To further validate the other utilities of our information-dense and omni-dimensional data, this section investigates its application in constructing a new supervised fine-tuning (SFT) dataset to enhance the spatial intelligence of existing T2I models.

**Additional SFT data construction.** SpatialT2I is constructed separately and has no overlap with our evaluation benchmark. The construction of SpatialT2I involves two stages:

- *Stage 1: Prompt and Omni-dimensional QAs generation.* This stage follows the same principles of our SpatialGenEval in Section 2.3. Totally, we obtain another 1,230 prompts and 12,300 QAs.
- *Stage 2: Rewrite prompt to obtain text-image pair.* We curated outputs from 14 top-performing T2I models with average scores above 50% (Table 2,10), along with their generation prompts. These are processed by a strong MLLM (*e.g.*, Gemini 2.5 Pro) to produce mildly rewritten prompts that better match the corresponding images, improving text-image consistency while preserving information density and all dimensions of spatial intelligence. The meta instruction of SpatialT2I construction is shown in the Appendix A.5. It is worth noting that data from "Design" scenes (130 prompts) are excluded due to low image quality. In total, we construct $(1230-130) \times 14 = 15,400$ image-text pairs from 22 scenes, forming the **SpatialT2I** dataset for the following SFT stage.

**Training details and SFT results.** We fine-tune the recent UniWorld-V1 (Lin et al., 2025), OmniGen2 (Wu et al., 2025b), and Stable Diffusion-XL (Rombach et al., 2022) based on their official settings and our SpatialT2I dataset. As shown in Table 6, the fine-tuned models consistently achieve better spatial abilities in SpatialGenEval. Finally, we select fine-tuned OmniGen2 as our final model,

and the qualitative comparisons with recent SOTA T2I models are presented in Figure 6. The generated images exhibit competitive results and more realistic effects.

**Ablation study of SpatialT2I and trend on data scaling.** To assess the impact of data quality and quantity in SpatialT2I, we conduct two experiments as follows. (1) We follow SPRIGHT (Chatterjee et al., 2024) and select three subsets arranged by increasing performance in Table 2, *i.e.*, Unipic-v2 (54.3), Bagel (57.0), and Qwen-Image (60.6). Each subset includes 1100 text-image pairs. As shown in Figure 7, fine-tuning on both the diffusion model (SD-XL) and non-diffusion model (OmniGen2) reveals that all subsets yield performance gains, and higher-scoring ones contribute more significantly. (2) We observe a data scaling trend in Figure 7, as performance consistently improves when increasing training data from 0%

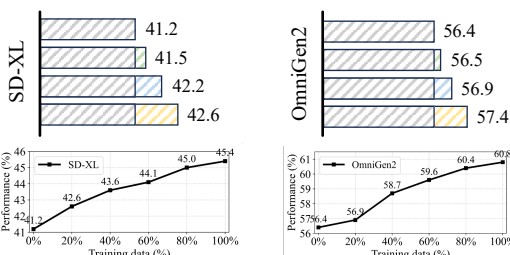

Figure 7: **(Top)** Ablations of the contribution of three different SpatialT2I subsets to fine-tuning performance. The selected subsets are arranged from top to bottom in order of their performance. **(Bottom)** Data scaling trend observed when incrementally adding better subsets to the training data.

to 100%, by progressively adding higher-scoring subsets. These findings reveal the value of exploring information-dense, spatial-aware data and suggest that further scaling is a promising direction.

## 5 RELATED WORK

**Text-to-Image models.** Text-to-image generation relies on several key architectural approaches. Diffusion models (Rombach et al., 2022) are the dominant paradigm, prized for a parallel generation process that yields high efficiency and global coherence. Innovations within this framework include scaling the generative backbone with transformer architectures (Black Forest Labs, 2024; Peebles & Xie, 2023) and improving semantic comprehension with powerful LLM text encoders (Hu et al., 2024). Beyond diffusion, autoregressive models (Wu et al., 2025b; Han et al., 2025; NextStep Team et al., 2025) regard image generation as a token-by-token process, a sequential nature that inherently models strong dependencies between visual tokens while offering finer compositional control. A recent trend is unified multimodal architectures (Deng et al., 2025; Chen et al., 2025b; Xie et al., 2024; Lin et al., 2025; Skywork Multimodality Team, 2025), which integrate visual understanding and generation into a single model for more holistic reasoning. Totally, all these diverse models have enabled models to excel at generating high-fidelity images with basic compositional elements.

**Text-to-Image benchmarks.** The evaluation of T2I models has co-evolved with their capabilities, leading to distinct categories of benchmarks. The first category targets foundational semantic alignment, verifying object presence and attribute binding with metrics like object detection (Ghosh et al., 2023; Huang et al., 2023). A more recent category addresses complex instruction following and relational understanding, using longer prompts and question-answering formats to assess multi-object relationships (Wei et al., 2025; Hu et al., 2024; Chang et al., 2025; Chatterjee et al., 2024). Furthermore, a growing number of specialized benchmarks have emerged to probe even higher-order capabilities, such as world knowledge (Niu et al., 2025), physical plausibility (Meng et al., 2024), broader reasoning skills (Chen et al., 2025a), and comprehensive quantitative understanding of T2I models' capabilities and risks (Lee et al., 2023). This evolution highlights a clear shift in research focus of T2I evaluation, from object-level fidelity towards scene-level compositional logic.

## 6 CONCLUSION

In this paper, we introduce SpatialGenEval, a benchmark that systematically evaluates the spatial intelligence of text-to-image (T2I) models. It is built upon a hierarchical framework and utilizes information-dense prompts to test model capabilities in scenarios of real-world complexity. Our extensive evaluation of current models reveals a stark performance disparity between basic object generation and advanced spatial tasks, pinpointing spatial reasoning as the primary bottleneck. Furthermore, by demonstrating the effectiveness of our supervised fine-tuning dataset, SpatialT2I, we validate a data-centric approach as a practical path toward resolving these shortcomings.

## ETHICS STATEMENT

This paper introduces SpatialGenEval, a benchmark for evaluating the spatial intelligence of text-to-image models. Its creation uses large multimodal models (MLLMs) with rigorous human oversight. All generated prompt and question-answer pairs are reviewed by human experts to ensure it is logical, neutral, and free of harmful or personally identifiable information. The benchmark is based on common, real-world scenes to ensure broad applicability. Our research aims to transparently identify current AI limitations, thereby fostering the development of more capable and reliable text-to-image generative models.

## REPRODUCIBILITY STATEMENT

To ensure reproducibility, this paper provides comprehensive methodological details, and all resources are made publicly available. The construction of the SpatialGenEval benchmark is detailed in Section 2 and Appendix A.2, A.4. Our experimental setup, including the models, judges, and evaluation protocol, is described in Section 3. The creation of the SpatialT2I dataset is outlined in Section 4 and Appendix A.5. The complete benchmark, dataset, and evaluation code are available to the research community for verification and future work.

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

# A  APPENDIX

## A.1  BENCHMARK STATISTICS AND ANALYSIS

This section provides a detailed statistical overview (Table 7) of our SpatialGenEval benchmark, followed by a discussion of the selection principles that ensure its diversity and comprehensiveness.

| Statistic | Number |
|---|---|
| Number of prompts | 1,230 |
| - max length (words) | 97 |
| - min length (words) | 46 |
| - average length (words) | 66 |
| Number of scenes | 25 (1,230) |
| - Nature | 7 (350, 28.5%) |
| - Indoor | 4 (200, 16.3%) |
| - Outdoor | 8 (400, 32.5%) |
| - Human | 3 (150, 12.2%) |
| - Design | 3 (130, 10.6%) |
| Number of questions | 12,300 |
| - Multi-choice QAs | 12,300 (100%) |
| Number of spatial domains | 4 |
| Number of spatial sub-domains | 10 |

Table 7: Statistics results of our SpatialGenEval benchmark.

**Scene selection and diversity.** As illustrated in Figure 8, our selection of 5 primary scenes (nature, indoor, outdoor, human, and design) and their corresponding 25 sub-scenes is designed to form the representative samples of real-world applications where spatial intelligence is crucial. This set is a practical balance between broad coverage and a manageable benchmark size. Moreover, *our focus on diverse, real-world scenes is also timely, developing in parallel with a similar emphasis from leading models like Qwen-Image* (Wu et al., 2025a) *and benchmarks like OneIG-Bench* (Chang et al., 2025). Specifically, the detailed scenes are as follows.

- Outdoor (32.5%). As the largest category, this focuses on complex, public human environments. It includes transportation hubs (Airport, Railway), recreational areas (Park, Zoo), and commercial/-cultural spaces (Shopping Mall, Art Gallery, Cafe, Library). These scenes challenge models with high-density object layouts, crowd dynamics, and understanding large-scale functional designs.

- Nature (28.5%). This category spans large-scale environments from natural landscapes (Forest, Mountain, Desert, Beach, Underwater) to human settlements (Cityspace, Village). These scenes test reasoning about organic layouts, vast scales, and perspective (*e.g.*, the relative size of mountains or the dense arrangement of trees).

- Indoor (16.3%). This category includes common, confined spaces where function dictates object placement. Scenes like Kitchen, Classroom, Living Room, and Office test a model's understanding of strict physical constraints, containment, and the functional relationships between objects (*e.g.*, a chair's position relative to a desk).

- Human (12.2%). This category centers on people and their interactions. Scenes like Sports, Human Activities, and Portraits require reasoning about body poses, relative positions between multiple people, and human-object interactions, which are critical for depicting action and social context.

- Design (10.6%). This category tests spatial intelligence in non-photorealistic and conceptual contexts. Scenes like Cartoon, Advertisement, and Story challenge a model to generalize beyond real-world physics, understanding instead principles of artistic composition, narrative flow, and symbolic spatial arrangements.

**Domain selection and diversity.** Our selection of 4 domains and 10 sub-domains is not a random list, but a structured framework based on the definition of spatial intelligence. Drawing from studies in cognitive science (Malanchini et al., 2020; Gupta et al., 2021; Ruan et al., 2025) and computer vision (Yang et al., 2025a; Stogiannidis et al., 2025; Gong et al., 2025; Yang et al., 2025b; Cai et al., 2025), spatial intelligence can be seen as a series of steps. The process starts with the basic

| Models | Year | Resolution | Source | URL |
|---|---|---|---|---|
| *1. Diffusion Generative Model* | | | | |
| SD-1.5 | 2021.12 | 1024×1024 | checkpoint | https://huggingface.co/stable-diffusion-v1-5/stable-diffusion-v1-5 |
| PixArt-alpha | 2023.10 | 1024×1024 | checkpoint | https://huggingface.co/PixArt-alpha/PixArt-XL-2-1024-MS |
| Playground-v2.5 | 2024.02 | 1024×1024 | checkpoint | https://huggingface.co/playgroundai/playground-v2.5-1024px-aesthetic |
| SD-XL | 2023.07 | 1024×1024 | checkpoint | https://huggingface.co/stabilityai/stable-diffusion-xl-base-1.0 |
| PixArt-sigma | 2024.04 | 1024×1024 | checkpoint | https://huggingface.co/PixArt-alpha/PixArt-Sigma-XL-2-1024-MS |
| SD-3-M | 2024.03 | 1024×1024 | checkpoint | https://huggingface.co/stabilityai/stable-diffusion-3-medium |
| SD-3.5-L | 2024.11 | 1024×1024 | checkpoint | https://www.modelscope.cn/models/AI-ModelScope/stable-diffusion-3.5-large |
| SANA 1.5 | 2025.01 | 1024×1024 | checkpoint | https://huggingface.co/Efficient-Large-Model/SANA1.5_4.8B_1024px_diffusers |
| FLUX.1-dev | 2024.12 | 1024×1024 | checkpoint | https://huggingface.co/black-forest-labs/FLUX.1-dev |
| FLUX.1-krea | 2025.07 | 1024×1024 | checkpoint | https://huggingface.co/black-forest-labs/FLUX.1-Krea-dev |
| Qwen-Image | 2025.08 | 1328×1328 | checkpoint | https://huggingface.co/Qwen/Qwen-Image |
| *2. AutoRegressive Generative Model* | | | | |
| OminiGen2 | 2025.06 | 1024×1024 | checkpoint | https://github.com/VectorSpaceLab/OmniGen2 |
| NextStep-1 | 2025.08 | 512×512 | checkpoint | https://github.com/stepfun-ai/NextStep-1 |
| Infinity | 2024.12 | 1024×1024 | checkpoint | https://github.com/FoundationVision/Infinity |
| *3. Unified Generative Model* | | | | |
| Show-o | 2024.08 | 512×512 | checkpoint | https://github.com/showlab/Show-o |
| Janus-Pro | 2025.01 | 384×384 | checkpoint | https://github.com/deepseek-ai/Janus |
| UniWorld-V1 | 2025.06 | 1024×1024 | checkpoint | https://github.com/PKU-YuanGroup/UniWorld-V1 |
| UniPic-v2 | 2025.08 | 512×384 | checkpoint | https://github.com/SkyworkAI/UniPic/tree/main/UniPic-2 |
| Bagel | 2025.05 | 1024×1024 | checkpoint | https://github.com/ByteDance-Seed/Bagel |
| *4. Closed-source Generative Model* | | | | |
| DALL-E-3 | 2023.10 | 1024×1024 | API | https://openai.com/zh-Hans-CN/index/dall-e-3 |
| GPT-Image-1 | 2025.04 | 1024×1024 | API | https://platform.openai.com |
| Nano Banana | 2025.08 | 1024×1024 | API | https://developers.googleblog.com/en/introducing-gemini-2-5-flash-image |
| Seed Dream 4.0 | 2025.08 | 1024×1024 | API | https://research.doubao.com/zh/seedream4_0 |

Table 8: The release time, resolution, model source, and URL of T2I models on our SpatialGenEval.

step of identifying what objects are present. It then moves on to perceiving their static properties and arrangements. The next step is to infer the relationships between them. The final stage is understanding the dynamics and causes of events. Our framework is designed to follow this logical progression, making it a powerful diagnostic tool.

**T2I model selection and diversity.** To provide a comprehensive overview of the current T2I landscape, we select 23 representative models, as detailed in Table 8. Our selection spans multiple architectural paradigms, including dominant diffusion models (Rombach et al., 2022; Chen et al., 2023; 2024b; Li et al., 2024; Xie et al., 2025; Black Forest Labs, 2024; Wu et al., 2025a), autoregressive models (Wu et al., 2025b; NextStep Team et al., 2025; Han et al., 2025), and emerging unified architectures (Xie et al., 2024; Chen et al., 2025b; Skywork Multimodality Team, 2025; Lin et al., 2025; Deng et al., 2025). Crucially, our evaluation includes both open-source checkpoint models and leading closed-source API systems, such as DALL-E-3 (Ramesh et al., 2021), GPT-Image-1 (OpenAI, 2024b), and Nano Banana (Google, 2025). This dual approach allows for a direct comparison between community-driven and commercial efforts, ensuring our benchmark's findings are robust and widely applicable across the entire T2I ecosystem.

**Question examples.** As the question examples shown in Table 9, our question design deliberately moves beyond simple checks to probe deeper and more complex spatial understanding as follows.

- For Spatial Foundation, instead of asking for the presence of a single object, we test comprehensive scene awareness by asking what else exists besides a given set of objects. For Attribute, we query for multi-attribute combinations rather than just single, isolated properties.

- For Spatial Perception, questions require models to determine an object's position or orientation from another object's viewpoint, not just a global third-person perspective. This tests a more sophisticated understanding of relative and absolute spatial arrangements.

- For Spatial Reasoning, our questions shift from qualitative judgments to quantitative analysis. For instance, Comparison tasks demand reasoning about how many times a quantity or by how much larger, going beyond simple "more/less" distinctions. We also probe 3D spatial understanding through Occlusion and physical distance-based Proximity.

- For Spatial Interaction, we focus on the dynamics between multiple objects and their effects. Questions target the causal outcomes of these interactions, demanding a level of reasoning far beyond the description of simple, isolated motions.

| Task | Question Examples |
|------|-------------------|
| Object | - Besides [object A] and [object B], what other objects are mentioned in the image?
- Which option can better describe [all objects] exist in the image? |
| Attribute | - Which option can better describe the attributes of the [object A] and [object B] in the image?
- What is the [attribute A], [attribute B], and [attribute C] of the [object] in the image? |
| Position | - From the third-person perspective, where is the [object] in the image?
- From the [object A]'s perspective, where is the [object] in the image? |
| Orientation | - From the third-person perspective, what is the facing orientation of the [object] in the image?
- From the [object A]'s perspective, what is the facing orientation of the [object B] in the image?
- From the third-person perspective, what are the [object A] and [object B] facing orientations in the image? |
| Layout | - How are the objects arranged in the image?
- How are the [object A] and [object B] arranged in the image?
- How are the [a group of object A] and [object B] arranged in the image? |
| Comparison | - How many times the quantity of [object A] that of [object B]?
- What is the size/height/numerical difference between the [object A], [object B], and [object C] in the image? |
| Proximity | - What is the closest object to the [object A] in the image?
- Which objects are [nearby object A] in the image? |
| Occlusion | - Is the [object A] fully visible in the image?
- What part of the [object A] is partially occluding the [object B] in the image? |
| Motion | - What is the motion interaction between the [object A] and [object B] in the image?
- What is the interaction between [object A], [object B], and [object C] in the image? |
| Causal | - What caused the flag to flutter in the wind in the image?
- What caused the floor to become illuminated in the image? |

Table 9: Question examples of all spatial sub-domains in SpatialGenEval.

## A.2 HUMAN ANNOTATION INTERFACE

To ensure the quality and accuracy of SpatialGenEval, we employ a rigorous two-stage human refinement process. This process involves five expert annotators and requires over 168 person-hours to complete. To maintain high inter-annotator agreement, all annotators first complete a calibration phase using a detailed guidebook. The details of the two stages are as follows. The human annotation interface is shown in Figure 9.

- *Prompt refinement.* In this stage, annotators revise the initial text prompts to improve clarity, fluency, and logical consistency. Specifically, they follow two strict guidelines: (a) simplifying vocabulary by replacing or removing uncommon words, and (b) verifying that each prompt comprehensively covers all 10 spatial sub-domains without omission.

- *QA refinement.* In the second stage, annotators review the generated question-answer pairs against the refined prompts. This refinement process adheres to three key principles: (a) ensuring each question precisely targets a specific spatial sub-domain, (b) removing any information from the question that could directly leak the answer, and (c) correcting phrasing inaccuracies, such as changing "in the prompt" to "in the scene" or "in the image". After refinement, each QA pair is independently validated by Qwen2.5-VL (Bai et al., 2025) for consistency evaluation.

## A.3 ADDITIONAL CLOSED-SOURCE EVALUATION

In line with current leading benchmarks (Wei et al., 2025; Li et al., 2025), we conduct a secondary evaluation using a powerful closed-source model, GPT-4o-250306, as an alternative evaluator. Table 10 presents the full results. The model rankings from GPT-4o are highly consistent with those from our primary evaluation using Qwen2.5-VL-72B. This consistency confirms the robustness of our main analysis, demonstrating that our conclusions are not dependent on a single evaluator.

## A.4 META INSTRUCTION OF SPATIALGENEVAL CONSTRUCTION

The construction of our benchmark and dataset follows a semi-automated pipeline, leveraging large multimodal models (MLLMs) with human oversight. In this section, we detail the core meta instructions used at each stage of the data generation process.

| Model | Size | Overall | Spatial Foundation | | Spatial Perception | | | Spatial Reasoning | | | Spatial Interaction | |
|---|---|---|---|---|---|---|---|---|---|---|---|---|
| | | | Object | Attribute | Position | Orientation | Layout | Comparison | Proximity | Occlusion | Motion | Causal |
| *1. Diffusion Generative Model* | | | | | | | | | | | | |
| SD-1.5 | 0.86B | 24.6 | 35.3 | 43.4 | 15.5 | 20.2 | 29.4 | 8.0 | 24.2 | 13.7 | 28.0 | 28.1 |
| PixArt-alpha | 0.6B | 35.1 | 44.2 | 58.8 | 28.2 | 39.9 | 38.4 | 11.4 | 35.0 | 17.9 | 35.5 | 42.1 |
| Playground-v2.5 | 3.5B | 38.6 | 54.6 | 64.2 | 29.8 | 41.0 | 42.5 | 12.3 | 39.7 | 19.4 | 37.6 | 44.8 |
| SD-XL | 6.5B | 38.6 | 55.1 | 60.8 | 28.0 | 39.6 | 45.4 | 13.3 | 40.5 | 21.6 | 39.3 | 42.2 |
| PixArt-sigma | 0.6B | 48.0 | 66.7 | 74.4 | 39.3 | 47.8 | 51.1 | 16.8 | 50.4 | 26.4 | 49.4 | 58.0 |
| SD-3-M | 2B | 51.1 | 74.6 | 76.2 | 42.8 | 49.8 | 57.5 | 20.3 | 56.2 | 22.8 | 52.0 | 58.5 |
| SD-3.5-L | 8.1B | 52.9 | 74.2 | 79.0 | 39.8 | 51.4 | 58.9 | 19.9 | 56.9 | 29.5 | 58.1 | 61.5 |
| SANA 1.5 | 4.8B | 53.0 | 71.5 | 78.0 | 43.3 | 51.5 | 57.2 | 18.6 | 56.7 | 27.9 | 58.3 | 66.7 |
| FLUX.1-dev | 12B | 54.3 | 74.6 | 78.3 | 44.8 | 56.4 | 60.5 | 20.3 | 59.7 | 27.6 | 60.2 | 60.7 |
| FLUX.1-krea | 12B | 58.0 | 79.5 | 79.8 | 48.9 | 55.9 | 63.8 | 20.7 | 63.7 | 29.8 | 65.6 | 71.9 |
| Qwen-Image | 20B | 60.8 | 78.3 | 83.7 | 49.3 | 59.3 | 66.7 | 21.7 | 66.4 | 36.0 | 72.5 | 74.1 |
| *2. AutoRegressive Generative Model* | | | | | | | | | | | | |
| OmniGen2 | 4B | 52.3 | 73.9 | 75.0 | 50.2 | 51.4 | 60.6 | 18.9 | 57.8 | 23.9 | 53.2 | 57.9 |
| NextStep-1 | 14B | 52.4 | 67.2 | 69.8 | 44.7 | 53.5 | 59.9 | 20.5 | 56.0 | 28.6 | 56.6 | 67.4 |
| Infinity | 8B | 54.6 | 74.9 | 75.7 | 50.0 | 56.3 | 62.4 | 20.0 | 58.9 | 28.2 | 56.0 | 64.1 |
| *3. Unified Generative Model* | | | | | | | | | | | | |
| Janus-Pro | 7B | 48.0 | 61.1 | 64.1 | 42.7 | 47.4 | 55.4 | 17.2 | 53.3 | 27.6 | 50.2 | 61.2 |
| Show-o | 1.3B | 48.9 | 63.6 | 69.1 | 44.7 | 48.0 | 56.3 | 18.8 | 51.7 | 24.7 | 51.4 | 60.3 |
| UniWorld-V1 | 12B | 50.9 | 72.6 | 71.6 | 45.9 | 52.4 | 59.8 | 18.7 | 54.6 | 22.8 | 54.6 | 56.0 |
| UniPic-v2 | 9B | 51.6 | 64.1 | 70.4 | 45.6 | 50.5 | 59.1 | 21.4 | 57.0 | 27.9 | 55.1 | 65.2 |
| Bagel | 7B | 56.6 | 76.7 | 79.8 | 45.9 | 55.7 | 62.4 | 20.6 | 62.1 | 31.3 | 64.2 | 67.6 |
| *4. Closed-Source Generative Model* | | | | | | | | | | | | |
| DALL-E-3 | - | 51.8 | 66.7 | 69.3 | 40.2 | 53.7 | 59.5 | 21.1 | 56.3 | 24.6 | 61.5 | 65.4 |
| GPT-Image-1 | - | 59.2 | 70.4 | 73.5 | 52.2 | 61.3 | 65.4 | 24.1 | 67.2 | 30.8 | 73.3 | 74.0 |
| Nano Banana | - | 61.6 | 68.0 | 74.5 | 56.3 | 62.0 | 69.0 | 26.4 | 70.7 | 36.1 | 74.6 | 78.3 |
| Seed Dream 4.0 | - | 62.1 | 66.5 | 77.6 | 52.2 | 62.3 | 68.6 | 26.4 | 70.3 | 38.0 | 77.1 | 81.5 |

Table 10: SpatialGenEval leaderboard based on GPT-4o.

**Stage 1: Meta instruction for prompt generation.** Following the key design principles of long, information-dense, and spatial-aware settings, we instruct the MLLM to synthesize a coherent scene description that incorporates all 10 spatial sub-domains based on the given scene as follows.

---

**Stage 1: Meta Instruction for Prompt Generation**

**[Task Description]**
I am creating a text-to-image evaluation benchmark to challenge the spatial intelligence of text-to-image models. You are an assistant tasked with generating 50 distinct and dynamic text-to-image prompts based on the provided scene. Each prompt MUST involve 10 types of spatial sub-domains as follows.

**[Scene]**
###Scene###

**[Definitions of 4 Spatial Primary-domains]**
4 primary spatial domains involve spatial foundation, spatial perception, spatial reasoning, and spatial interaction.

**[Definitions of 10 Spatial Sub-domains]**
10 spatial sub-domains involve object, attribute, position, orientation, layout, comparison, proximity, occlusion, motion interaction, and causal motion interaction.

**[Instructions]**
1. The prompt should clearly describe all 10 spatial sub-domains around 60 words.
2. The generated 50 prompts should be distinct and involve various cases.

**[Output Format]**
Please output your response in valid JSON format as follows.
{scene: ###Scene###, prompt: ###The generated prompt 1###},
{scene: ###Scene###, prompt: ###The generated prompt 2###},
{......},
{scene: ###Scene###, prompt: ###The generated prompt 50###}

---

**Stage 2: Meta instruction for QAs generation.** For each generated prompt, we instruct the MLLM (*i.e.*, Gemini 2.5 Pro (Comanici et al., 2025)) to create 10 corresponding multiple-choice questions, each targeting one of the 10 spatial sub-domains to enable omni-dimensional evaluations as follows.

---

**Stage 2: Meta Instruction for Question-Answer Pairs Generation**

**[Task Description]**
I am creating a text-to-image evaluation benchmark to challenge the spatial understanding ability of text-to-image models. You are an assistant tasked with generating **10 multiple-choice question-answers** based on the given **text-to-image prompt**. Each question MUST involve one of the following 10 spatial sub-domains. Please avoid including any question that introduces irrelevant or mythological information not present in the prompt.

**[Definitions of 4 spatial primary-domains]**
4 spatial primary-domains involve spatial foundation, spatial perception, spatial reasoning, and spatial interaction.

**[Definitions of 10 spatial sub-domains]**
10 spatial sub-domains involve object, attribute, position, orientation, layout, comparison, proximity, occlusion, motion interaction, and causal motion interaction.

**[Output Format]**
Based on the following ###prompt### to generate 10 multiple-choice question-answers and fill in the following "questions" and "answers" fields. The questions MUST be in the same order as the "question type" field. Please avoid including any question that introduces irrelevant or mythological information not present in the prompt.
{id: ###id###, scene: ###scene###, prompt: ###prompt###, question type: ###question type###, questions: [###question 1###, ###question-2###, ..., ###question-10###], answers: [###answer-1###, ###answer-2###, ..., ###answer-10###]}

---

**Stage 3: Meta instruction of MLLM evaluation.** To evaluate a generated image, we provide an image and its 10 corresponding multiple-choice questions to an MLLM evaluator. The instruction explicitly forbids the use of external knowledge and enforces a visually-grounded answering process. The meta instruction is as follows.

---

**Stage 3: MLLM Evaluation Instruction**

**[Task Description]**
You are tasked with carefully examining the provided image and answering the following 10 multiple-choice questions. You MUST ONLY rely on the provided image to answer the questions. DO NOT use any external resources like world knowledge or external information beyond the provided image.

**[Multiple-Choice Questions]**
###Multiple-Choice Questions###

**[Instructions]**
1. Answer these 10 questions on a separate 10 lines, beginning with the correct choice option (A/B/C/D/E) and followed by a detailed reason (in the same line as the answer).
2. Maintain the exact order of the questions in your answers.
3. Provide only one answer per question.
4. Each answer must be on its own line.
5. Ensure the index of answers matches the index of questions.
6. Select the option "E: None" when the image can not answer the question.

---

## A.5 META INSTRUCTION OF SPATIALT2I CONSTRUCTION

**Meta instruction for rewritten prompt generation of SpatialT2I.** To create the SpatialT2I dataset, we use an MLLM (*i.e.*, GPT-4o (OpenAI, 2024a)) to rewrite the original prompts to accurately describe the content of the generated images, thereby correcting spatial failures in text-image alignment. The core instruction is as follows.

---

**Meta Instruction for SpatialT2I Construction (Rewrite Prompt based on MLLM)**

**[Task Description]**
You are an expert AI assistant specializing in image prompt analysis and refinement. Your task is to analyze a generated image and its corresponding metadata to rewrite the original text-to-image prompt. The goal is to create a new prompt that accurately describes the spatial foundation, perception, reasoning, and interaction of the generated image.

**[Input Data]**
You will receive two primary inputs for each task:
1. A generated image: This is the visual ground truth and your primary source of information. Your final rewritten prompt must be a faithful description of this image.
2. A JSON input containing the following 7 keys:
{id: ###id###, scene: ###scene###, prompt: ###prompt###, question ###questions###, ground-truth answers: ###answers###, image path: ###image path###, answers from generated image: ###model preds###}

**[Instructions]**
Your process is to rewrite the prompt fully based on the real generated image as follows.
**Step 1:** Pinpoint discrepancies: Your primary task is to systematically compare the "ground-truth answers" with the "answers from generated image".

- If the answers match: This means the aspect of the image described by the question is generated correctly, and the corresponding part of the original prompt is accurate.

- If the answers DO NOT match: This is a critical signal. It tells you exactly where the generated image failed to follow the original prompt's instructions. The "answers from generated image" becomes your source of truth for this specific detail.

**Step 2:** Synthesize the rewritten prompt: Based on your analysis from Step 1 and direct observation of the image, you will construct a new, cohesive prompt.

- Embrace the reality: Your new prompt must be built from the facts presented in the "answers based on generated image" and confirmed by your own visual inspection.

- Integrate corrections holistically: Do not simply swap words. Weave the corrected details (object order, actions, attributes, *etc.*) into a new, flowing, and descriptive sentence.

- Be specific: Use precise and descriptive language that captures the nuances of the generated image. If the image shows a "rusty blue robot" instead of just a "rusty robot", your new prompt must include "blue."

- Verify everything: Before finalizing, re-read your rewrite prompt and ensure every single clause is verifiably true by looking at the provided image.

**[Output Format]**
Please ONLY output your response in a valid JSON format as follows.
{id: ###id###, scene: ###scene###, image path: ###image path###, original prompt: ###prompt###, rewrite prompt: ###rewrite prompt###}

---

## A.6 META INSTRUCTION OF PROMPT REWRITING

**Meta instruction for prompt rewriting.** We instruct Gemini 2.5 Pro (Comanici et al., 2025) to rewrite the original prompts with the specific goal of making our defined 10 spatial dimensions more explicit and unambiguous. The meta instruction is as follows.

| Model | Overall | Spatial Foundation | | Spatial Perception | | | Spatial Reasoning | | | Spatial Interaction | |
|---|---|---|---|---|---|---|---|---|---|---|---|
| | | Object | Attribute | Position | Orientation | Layout | Comparison | Proximity | Occlusion | Motion | Causal |
| SD-3.5-L | 54.0 | 52.4 | 72.0 | 44.7 | 52.0 | 62.7 | 25.4 | 61.3 | 27.4 | 69.4 | 72.6 |
| + Rewriting | 56.3 | 53.6 | 73.5 | 49.4 | 52.4 | 65.0 | 27.7 | 65.5 | 27.6 | 72.8 | 75.7 |
| OmniGen2 | 56.4 | 51.5 | 73.6 | 55.9 | 55.5 | 65.4 | 26.0 | 64.2 | 27.3 | 72.0 | 72.6 |
| + Rewriting | 58.5 | 53.8 | 75.6 | 60.1 | 55.5 | 68.4 | 30.5 | 63.8 | 27.7 | 75.0 | 75.0 |
| UniWorld-V1 | 54.2 | 46.8 | 71.3 | 50.1 | 53.1 | 64.0 | 26.1 | 62.0 | 26.8 | 69.6 | 72.4 |
| + Rewriting | 55.9 | 47.6 | 72.1 | 53.8 | 53.4 | 66.0 | 29.1 | 64.8 | 26.9 | 71.5 | 73.8 |
| Qwen-Image | 60.6 | 61.0 | 77.2 | 55.6 | 56.7 | 69.7 | 28.6 | 67.7 | 30.8 | 78.1 | 80.2 |
| + Rewriting | 61.7 | 62.8 | 80.8 | 57.6 | 56.4 | 70.2 | 29.7 | 68.6 | 30.4 | 79.0 | 81.1 |

Table 11: Quantitative results of recent T2I models based on prompt rewriting.

---

**Meta Instruction of Prompt Rewriting to Improve Spatial Intelligence**

**[Task Description]**
You are an expert Text-to-Image prompt rewriter. You will receive a long, information-dense text-to-image prompt designed to evaluate the spatial intelligence of generative models. Your task is to rewrite this prompt to accurately and clearly describe its contents, attributes, spatial perception, spatial reasoning, and spatial interaction. The ultimate goal is to make the rewritten prompt suitable for downstream text-to-image generation.

**[Input Data]**
A JSON input containing the following 6 keys:
{id: ###id###, scene: ###scene###, prompt: ###prompt###, question type: ###question type###, question: ###questions###, ground-truth answers: ###answers###}

**[Instructions]**
**Step 1:** Analyze Input: Carefully examine all keys in the input JSON, especially the prompt, questions, and ground-truth answers.
**Step 2:** Deconstruct the Scene: Systematically go through each of the 10 question-answer pairs. Each answer provides a non-negotiable fact about the scene's final state (*e.g.*, what objects exist, their attributes, exact positions, layout, relative sizes, occlusion state, and the result of any interactions).
**Step 3:** Synthesize the Rewritten Prompt: Construct a clear prompt by integrating all 10 facts you confirmed in the previous step. Start with the foundational elements and layer in the perceptual, reasoning, and interactional details.
**Step 4:** Review your rewritten prompt to ensure it is a single, clear paragraph.

**[Output Format]**
Please ONLY output your response in a valid JSON format as follows.
{id: ###id###, scene: ###scene###, question type: ###question type###, original prompt: ###prompt###, rewrite prompt: ###rewrite prompt###}

---

## A.7   IMPACT OF PROMPT REWRITING TO IMPROVE SPATIAL INTELLIGENCE

To explore other solutions to improve the spatial intelligence of T2I models, we explore another potential method: prompt rewriting. We instruct Gemini 2.5 Pro (Comanici et al., 2025) to rewrite the original prompts with the specific goal of making our defined 10 spatial dimensions more explicit and unambiguous. These enhanced prompts are then sent to the text-to-image models (across diffusion-based, autoregressive-based, and unified-based) for evaluation. The results in Table 11 show that prompt rewriting is a viable strategy for improving model performance. The detailed breakdown reveals three key insights:

- *Enhanced prompt decomposition is a valuable path for T2I models.* The results show that all models benefit from rewriting, confirming that a model's core ability to deconstruct complex prompts is a critical bottleneck. Notably, the improvement is more pronounced for models that initially struggle with text reasoning (*e.g.*, SD-3.5-L shows a +2.3% gain), suggesting that improving this native capability is a promising research direction.

- *Rewriting achieves greater gains on explicit spatial relationships.* Rewriting proves highly effective at resolving textual ambiguity in categories like Position, Comparison, and Layout. This leads

to substantial score increases (*e.g.*, +4.5% in Comparison for OmniGen2, +4.7% in Position for SD-3.5-L), demonstrating its ability to clarify relational instructions.

- *Minimal impact on implicit visual reasoning.* Conversely, rewriting offers little benefit for complex visual reasoning tasks like Occlusion and Orientation. This indicates the failure stems not from text comprehension, but from a core lack of 3D and physical reasoning in the generator. Such challenges require solutions beyond prompt engineering, such as specialized fine-tuning or unified-based design for joint optimization.

### A.8   The Use of LLMs

This work utilizes several multimodal large language models (Comanici et al., 2025; OpenAI, 2024a; Bai et al., 2025) for benchmark construction, SFT data construction, and improving paper writing. We transparently document their roles, and our rationale for choosing them is as follows.

- **Gemini 2.5 Pro** (Comanici et al., 2025). We leverage Gemini 2.5 Pro for the initial data generation phase of SpatialGenEval and prompt rewriting for improvement, including (1) information-dense prompt generation, (2) omni-dimensional QA pairs generation, and (3) prompt rewriting to search for enhanced text understanding for generation. In preliminary comparisons against other MLLMs such as GPT-4o (OpenAI, 2024a), Qwen2.5-VL-72B (Bai et al., 2025), DeepSeek-V3 (DeepSeek-AI Team. et al., 2025), and Gemini-1.5-Pro (Gemini Team. et al., 2024), our human evaluations find that Gemini 2.5 Pro demonstrates a superior creative capability for generating novel, complex, and diverse content from scratch. This observation aligns well with the recent performance comparisons of MLLMs (OpenCompass, 2023).

- **Qwen2.5-VL-72B** (Bai et al., 2025). We employ Qwen2.5-VL-72B as the primary evaluator (judge), which is renowned for its strong open-source multimodal understanding capabilities. This choice ensures the long-term reproducibility of our evaluation results and validates our findings independently of closed-source APIs.

- **GPT-4o** (OpenAI, 2024a). GPT-4o serves two critical roles in our pipeline: (1) acting as the secondary closed-source evaluator (judge) for our benchmark, and (2) performing the prompt rewriting for the SpatialT2I dataset. We select it due to its widespread adoption, consistently stable multimodal understanding, and superior inference speed, which are crucial for our large-scale evaluation and data refinement tasks.

### A.9   Discussion

#### A.9.1   Discussion about the choice of Gemini 2.5 Pro to create T2I prompts

The choice of Gemini 2.5 Pro stems from its strong creative ability (less repetition than others across 50 prompts within the same scene). Specifically, the construction of SpatialGenEval benchmark concentrates on two abilities as follows.

- *Strong instruction following ability*: As stated in Section 2.3, the prompt generation process must strictly follow the instructions to generate each text-to-image prompt based on a given scene (*e.g.*, classroom) while seamlessly integrating all 10 pre-defined spatial sub-domains.

- *Strong creative ability (less repetition)*: As stated in Appendix A.4, we instruct the model to generate 50 distinct text-to-image prompts all at once. This requires a strong creative ability to avoid generating repetitive outputs.

During the implementation, we conducted a controlled experiment where we instructed three top-performing MLLMs (Gemini 2.5 Pro, GPT-4o, and Qwen2.5-VL-72B) for prompt generation. We found that all three models demonstrate strong instruction following ability, but Gemini 2.5 Pro outperformed the others in terms of creative ability. This observation stems from: (1) consistent decision from all five human annotators, and (2) Gemini 2.5 Pro shows lower average sentence similarity (Reimers & Gurevych, 2019) across all text-to-image prompts within the same scene, i.e., Gemini 2.5 Pro (0.4938) < GPT-4o (0.5125) < Qwen2.5-VL-72B (0.5548).

| Model | Spatial Foundation | Spatial Perception | Spatial Reasoning | Spatial Interaction | Overall |
|---|---|---|---|---|---|
| No Image Input | 7.3 | 19.2 | 13.5 | 28.1 | 16.9 |
| Random Choice | 19.7 | 19.7 | 19.8 | 20.0 | 19.8 |
| Qwen-Image | 69.1 | 60.7 | 42.4 | 79.2 | 60.6 |

Table 12: Analysis of the potential visual bias from MLLM predictions.

### A.9.2 DISCUSSION ABOUT THE POTENTIAL BIAS FROM MLLM PREDICTIONS

To mitigate the potential biases from model predictions (*e.g.*, which favors reliance on implicit world knowledge to answer, or other aesthetic and stylistic preferences), our evaluation protocol is robust with three safeguards.

- *No external knowledge instruction.* The MLLM is explicitly instructed: *DO NOT use any external resources like world knowledge.* This discourages guessing based on prior knowledge.
- *Refuse to answer (add another "E: None" option).* The inclusion of the "E: None option" is crucial. If the image does not provide the necessary visual evidence to answer the question, the evaluator is instructed to select "E: None", preventing forced-choice guessing.
- *Majority voting.* We employ a 5-round voting mechanism, where a response is only considered correct if the ground-truth answer is selected in at least 4 of the 5 rounds. This enhances evaluation stability and reduces the impact of random inference.
- *Questions are designed to be aesthetic-agnostic.* The task of MLLM evaluator is to perform direct Visual Question Answering (VQA) based on factual spatial correctness (*e.g.*, "How are the children arranged around the storyteller?"). The questions have no aesthetic or stylistic preference.

Finally, to empirically validate that our questions are visually grounded, we conduct a new ablation study: *evaluating the questions without any image input*. The results in Table 12 show that the overall accuracy drops to 16.9%, lower than the random guess accuracy of 19.8%. These results indicate that although the model may exhibit some slight biases (*e.g.*, 28.1% vs. 20.0% in Spatial Interaction), their impact remains minimal. This result also strongly indicates that without visual context, the model cannot deduce the correct answers and often correctly refuses to answer (by selecting "E: None", which is marked as an incorrect answer since the ground truth is A-D).

### A.9.3 DISCUSSION ABOUT THE RELIANCE OF MLLM FOR EVALUATION.

Although current MLLMs remain working on handling highly complex spatial reasoning (Liu et al., 2023; Chen et al., 2024a; Zhang et al., 2024), we would like to clarify that current leading MLLMs are *suitable* and *capable* in evaluating the questions in our SpatialGenEval benchmark, based on three key aspects:

- *Design philosophy: The primary burden of spatial intelligence of our SpatialGenEval lies in the generative T2I model, not the VLM evaluator.* The T2I model must integrate the long, information-dense prompt to create an image with all defined spatial relationships. In contrast, the MLLM's task is simplified to simple visual checks based on the direct visual evidence. This design choice decouples the complex reasoning required for generation from the simpler verification required for evaluation.
- *Problem difficulty: Unlike benchmarks designed to push the boundaries of MLLM spatial understanding, SpatialGenEval focuses on evaluating generative tasks and consists of simpler QAs.* Unlike recent spatial intelligence benchmarks such as VSI-Bench (Yang et al., 2025a), MMSI-Bench (Yang et al., 2025b), and Space-10 (Gong et al., 2025), which target understanding task and focus on more abstract or complex long-range reasoning (*e.g.*, estimating real-world distances or path planning), SpatialGenEval is targeted to generative task and concentrates on evaluating more simpler visual relationships like relative position, layout, occlusion, and interaction. These tasks can be verified directly from the image's visual content, making them more suitable for current MLLMs.
- *Human alignment: Current leading MLLMs align well with human evaluation in our Spatial-GenEval.* Our human alignment study in the Section 3 provides direct numerical validation that current MLLMs are capable of handling our designed questions. The results reveal an exceptionally high correlation between the MLLM and human evaluations.

## A.10 LIMITATIONS

While our work establishes a new benchmark for evaluating spatial intelligence, we acknowledge two limitations. (1) *Scale concern*: The construction of SpatialGenEval relies on a semi-automated, human-in-the-loop pipeline. While this ensures high data quality, the process is labor-intensive and presents challenges for scaling the benchmark to an even larger size or to new domains. (2) *Scope expansion concern*: Our hierarchical framework of 10 spatial sub-domains, while comprehensive, is an abstraction of the near-infinite complexity of real-world spatial phenomena. More nuanced or dynamic spatial effects (Chen et al., 2024d; Chang, 2025), such as fluid dynamics, complex deformations, or multi-agent predictive interactions, are worth future exploration.

## A.11 FUTURE WORK

Our work encourages a shift from evaluating simple fidelity to complex compositional logic and opens several avenues for future research. (1) *Dataset-level extension*: Our key principles of information-dense prompts and omni-dimensional evaluation can be extended to other generative capabilities, such as stylistic control, textual rendering, and world knowledge. Extending this framework to text-to-video is also a crucial step for assessing spatio-temporal reasoning. (2) *Object-level extension*: Leveraging information-dense settings by increasing object number is a promising approach to improve fine-grained perception and generation (Li et al., 2025; Chatterjee et al., 2024). (3) *Post-training strategies*: Beyond simple fine-tuning, SpatialT2I enables exploring more advanced data-centric strategies. This includes curriculum learning (from simple to complex spatial tasks (Croitoru et al., 2025; Xiong et al., 2025)) and reinforcement learning (Wang et al., 2025; Liu et al., 2025; Chu et al., 2025a; Li et al., 2026; Dai et al., 2026) from MLLM feedback to further boost model performance.

## A.12 BROADER IMPACT

(1) *Creative tools*: Improving spatial intelligence can lead to more useful tools for artists, architects, and designers by enabling precise control over object placement and scene layout. (2) *Embodied AI*: Models with a stronger grasp of spatial relationships are a crucial step towards advancing AI agents for robotics and embodied intelligence (Duan et al., 2022), which must understand and interact with the physical world.

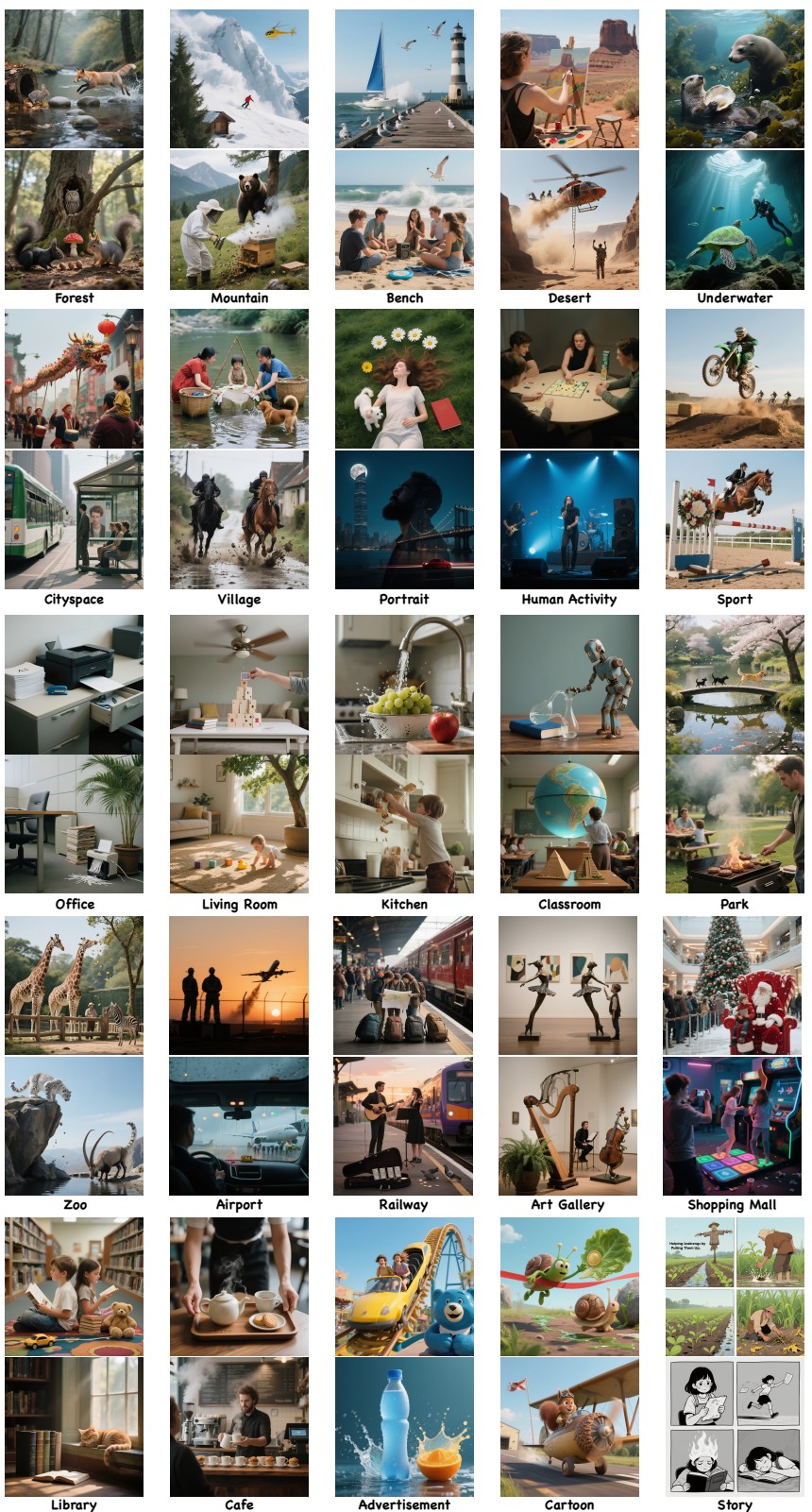

Figure 8: Samples of generated results from all selected 25 scenes in SpatialGenEval.

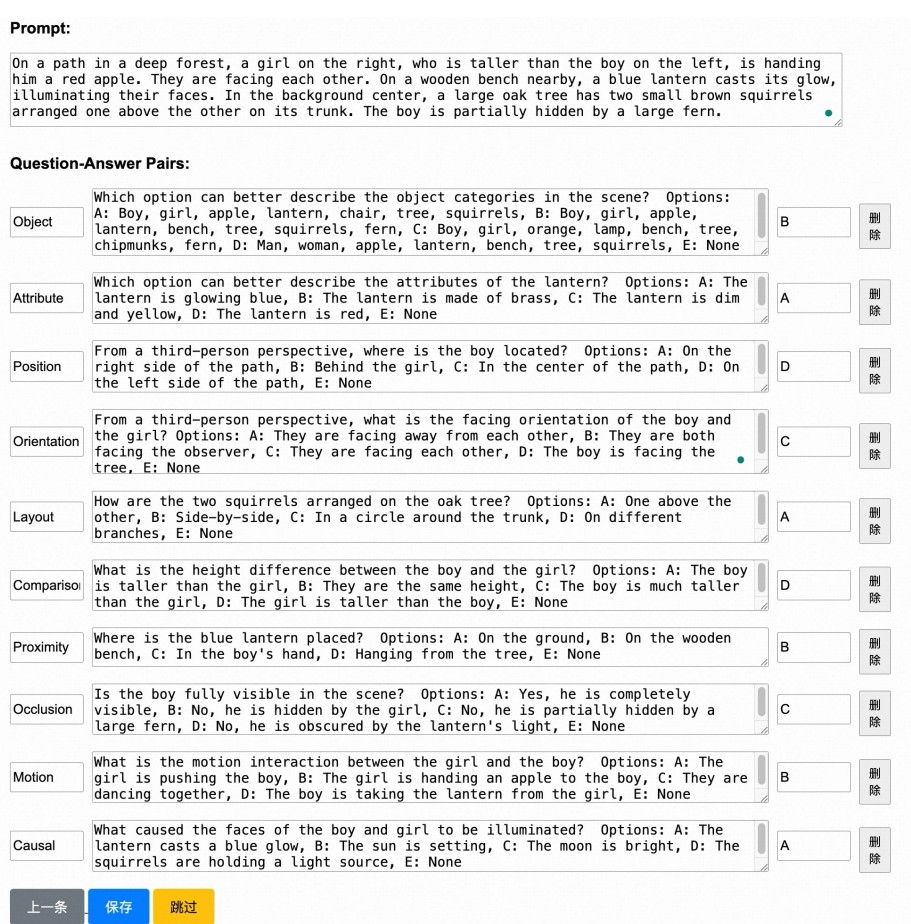

Figure 9: Illustration depicting the human annotation interface, where the experts are presented with the prompt and corresponding question-answer pairs for final refinement.

**Prompt**: A double exposure portrait features a woman with dark hair in profile, facing left. Her silhouette is filled with a landscape: on the right, the sun peeks over a large mountain. On the left, two bare trees stand in a river, the right one taller. Five birds fly toward the mountain in a V formation, and bright sunlight casts sharp shadows of the trees.

**S1: Spatial Foundation: Object**

Q1: Besides a woman, trees, and birds, what other major natural features are mentioned as being part of the landscape?

A: A mountain and a river
B: A forest and a lake
C: A desert and a moon
D: A valley and clouds          E: None

**S2: Spatial Foundation: Attribution**

Q2: What are the described attributes of the woman's hair and the trees?

A: Blonde hair, leafy trees          B: Dark hair, bare trees
C: Red hair, tall trees          D: Dark hair, green trees
E: None

**S3: Spatial Position**

Q3: Where are the two bare trees located within the landscape inside the silhouette?

A: On the right side, on the mountain
B: In the center, near the birds
C: On the left side, in a river
D: On the right side, in a river
E: None

**S4: Spatial Orientation**

Q4: What is the facing orientation of the woman in the portrait?
A: She is facing forward, looking at the viewer
B: She is in profile, facing right
C: Her back is to the viewer
D: She is in profile, facing left          E: None

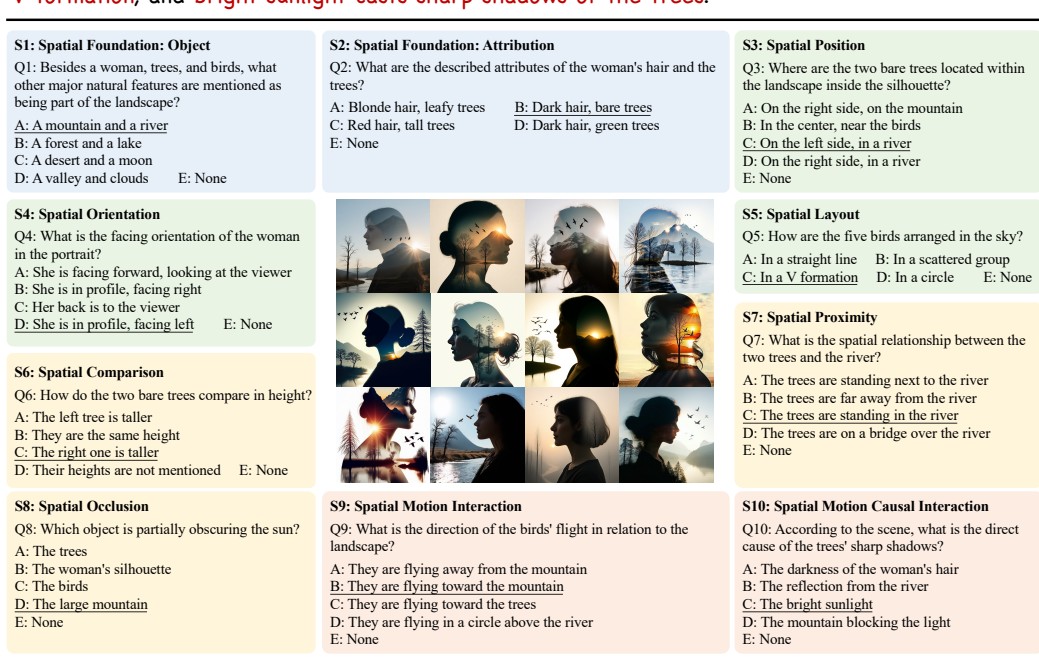

**S5: Spatial Layout**

Q5: How are the five birds arranged in the sky?
A: In a straight line          B: In a scattered group
C: In a V formation          D: In a circle          E: None

**S7: Spatial Proximity**

Q7: What is the spatial relationship between the two trees and the river?
A: The trees are standing next to the river
B: The trees are far away from the river
C: The trees are standing in the river
D: The trees are on a bridge over the river
E: None

**S6: Spatial Comparison**

Q6: How do the two bare trees compare in height?
A: The left tree is taller
B: They are the same height
C: The right one is taller
D: Their heights are not mentioned          E: None

**S8: Spatial Occlusion**

Q8: Which object is partially obscuring the sun?
A: The trees
B: The woman's silhouette
C: The birds
D: The large mountain
E: None

**S9: Spatial Motion Interaction**

Q9: What is the direction of the birds' flight in relation to the landscape?
A: They are flying away from the mountain
B: They are flying toward the mountain
C: They are flying toward the trees
D: They are flying in a circle above the river
E: None

**S10: Spatial Motion Causal Interaction**

Q10: According to the scene, what is the direct cause of the trees' sharp shadows?
A: The darkness of the woman's hair
B: The reflection from the river
C: The bright sunlight
D: The mountain blocking the light
E: None

Figure 10: A data sample from the SpatialGenEval benchmark, where the generated image is challenged by 10 question-answer pairs across 4 key dimensions: Spatial Foundation (S1, S2), Spatial Perception (S3-S5), Spatial Reasoning (S6-S8), and Spatial Interaction (S9, S10). The generated images are from 12 top-performing T2I models from top-left to bottom-right, *i.e.*, Qwen-Image (Wu et al., 2025a), GPT-4o (OpenAI, 2024b), Flux.1-krea (Black Forest Labs, 2024), Infinity Han et al. (2025), Bagel (Deng et al., 2025), Flux.1-dev (Black Forest Labs, 2024), OmniGen2 (Wu et al., 2025b), NextStep-1 (NextStep Team et al., 2025), DALL-E-3 (Ramesh et al., 2021), SD-3-M (Rombach et al., 2022), UniWorld-V1 (Lin et al., 2025), SD-3.5-L (Rombach et al., 2022).

