# OpenReview forum: "Everything in Its Place: Benchmarking Spatial Intelligence of Text-to-Image Models"
_ICLR.cc/2026/Conference — ICLR 2026 Poster_

### Official Review · Reviewer_awqq · 2025-10-30

**Soundness:** 3
**Presentation:** 4
**Contribution:** 4
**Rating:** 6
**Confidence:** 5

**Summary:**

The paper introduces SpatialGenEval, a new benchmark designed to systematically evaluate the spatial intelligence of T2I models. SpatialGenEval consists of 1,230 long, information-dense prompts across 25 real-world scenes  and 10 spatial sub-domains. SpatialGenEval moves beyond existing benchmarks by creating information dense prompts and evaluate 21 state-ofthe-art models. Evaluation reveals that higher-order spatial reasoning remains a primary bottleneck in existing models. Further, SpatialT2I dataset is generated, which consists of 15,400 text-image pairs, fine-tuning on which leads to performance improvement in models such as Uniworld-V1 and OmniGen2.

**Strengths:**

Having worked on spatial relationships in T2I models, I thank the authors for working on a very timely problem and doing a good job of it!

1. The motivation is outlined very well and the paper is easy to follow with supporting examples for easier readability.
2. Existing benchmarks, as rightly pointed by the paper, only present simpler prompts. SpatialGenEval does a comprehensive job of covering 10 spatial constraints, enabling deeper probing of models.
3. Holistic evaluation of multiple models, both open and closed-source along with difference architectures such as diffusion / AR models.
4. While the prompts and corresponding questions are generated from VLM's, a subsequent round of human validation is performed, which ensures a higher quality and avoids potential bias emanating from a proprietary LLM.

**Weaknesses:**

1. Reliance on VLM for evaluations - This is the major weakness of the current work. Generated images are evaluated with VLM's which themselves struggle with spatial reasoning. For example, a question such as :
 "Based on the toddler's movement, what is the toddler facing towards?" -- A: The ficus tree B: Away from the rubber duck C: The colorful blocks D: The window E: None ; will be difficult for most VLMs to answer as well. Despite Table 3 and 4, where meta comparisons to other benchmarks are shown, performing an alignment with human judgement will solidify the findings.

2. What is the design choice behind having all prompts having all the 10 spatial constraints? Incremental complexity would paint a better picture of where models fail; additionally since all these constraints are essentially "prompts"; some of these shortcomings might just be text encoder shortcomings instead of the core image generator blocks. This also increases complexity on the VLM's to "generate all 10 QA pairs at once".

3. While the intention is correct, some prompts/questions are ambiguous; for example in Spatial Causal Interaction - "Under a spotlight, a large opera singer in a tuxedo stands center stage. <···>. The sound wave hits the glass, causing it to shatter" - Since its an image, the glass can be shattered by a multiple number of possibilities; such as a rock thrown by someone in the crowd -- How are these cases handled?

4. The SFT contribution seems like an after thought, some more details will be helpful; for example: was the text encoder fine-tuned as part of the experiments? Please refer to SPRIGHT (https://arxiv.org/abs/2404.01197) where authors discuss useful training details for improving spatial relationships in T2I models and covers multiple related data-centric nuances mentioned about in the current work such as relative sizes, spatial proximity etc.

I will increase my score if the above points (especially 1) are addressed.

**Questions:**

Please refer to weaknesses.

---

> ### Author Response · Authors · 2025-11-21
> **Response to Reviewer#4 awqq (Part 1 of 3)**
>
> We are deeply grateful for your thoughtful and encouraging review. Your expertise in spatial relationships of T2I models is evident in your insightful suggestions to improve our paper. Below, we have carefully considered every suggestion and provided our point-by-point responses. Hope our response can address your concerns.
>
> > **Q1: Concern about the reliance of VLM for evaluation.**
>
> A1: Thank you for raising this critical point. We acknowledge the known limitations of VLMs in handling highly complex spatial reasoning (particularly in 3D spatial relations), just like the provided references. However, we would like to clarify that current leading VLMs are **suitable** and **capable** for evaluating the questions in our SpatialGenEval benchmark, based on three key aspects:
> 1. **Design philosophy: The primary burden of spatial intelligence of our SpatialGenEval lies in the generative T2I model, not the VLM evaluator.** The T2I model must integrate the long, information-dense prompt to create an image with all defined spatial relationships. In contrast, the VLM's task is simplified to simple visual checks based on the direct visual evidence. **This design choice decouples the complex reasoning required for generation from the simpler verification required for evaluation.**
> 2. **Problem difficulty: Unlike benchmarks designed to push the boundaries of VLM spatial understanding, SpatialGenEval focuses on evaluating generative task and consists of simpler QAs.** Unlike recent spatial intelligence benchmarks such as VSI-Bench [1], MMSI-Bench [2], and Space-10 [3], which target *understanding* task and focus on more abstract or complex long-range reasoning (e.g., estimating real-world distances or path planning), SpatialGenEval is targeted to *generative* task and concentrates on evaluating more simpler visual relationships like relative position, layout, occlusion, and interaction. These tasks can be verified directly from the image's visual content, making them more suitable for current VLMs.
> 3. **Human alignment: Current leading VLMs align well with human evaluation in our SpatialGenEval.** Our new human alignment study provides direct numerical validation that current VLMs are capable of handling our designed questions. The results reveal an exceptionally high correlation between the VLM and human evaluations, i.e., Qwen2.5-VL-72B (80.4%) and GPT-4o (78.8%). The detailed settings and results of our human alignment study are as follows.
>    - **Settings of human alignment study**. Following our VLM evaluation setup (\#Line 315-323), we randomly select 8 images from each of the 25 scenes, resulting in a total of 200 images. Five human annotators work independently to *select the best option* based solely on the *given image* and *10 questions*, with no access to the original text-to-image prompt to prevent "leakage". We follow the recent T2I benchmark [4] and adopt the *balanced accuracy* [5] as the overall human alignment score for alignment evaluations. All human annotators are experts in text-to-image generation, ensuring the quality and validity of the annotations.
>    - **Results of human alignment study**. The human alignment scores across 10 sub-domains are presented in the table below. We find that: (1) Overall alignment: The three selected models all achieve high scores. Gemini 2.5 Pro ranks the highest, followed by Qwen2.5-VL-72B and GPT-4o. (2) Alignment across sub-domains: We find that the human alignment correlates with the difficulty of each sub-domain. **The alignment is highest on simpler dimensions like Spatial Foundation and Perception, and lowest on the Spatial Reasoning sub-domain. Despite this, the alignment score still nears 80%**, validating the effectiveness of our chosen open-source and closed-source models as evaluators in SpatialGenEval.
>
> |VLMs|Overall|Object|Attribute|Position|Orientation| Layout | Comparison | Proximity | Occlusion | Motion | Causal |
> |:---|:---:|:---:|:---:|:---:|:---:| :---: | :---: | :---: | :---: | :---: | :---: |
> |Qwen2.5-VL-72B|80.4 | 84.5 |89.5|82.5|77.5|69.5|80.0|71.0|69.0|  84.0  |  85.5  |
> |GPT-4o|  78.8 |80.5|85.0|81.0|78.5|70.5|79.5|72.0|66.0|  80.5  |86.0|
> |Gemini 2.5 Pro|84.2|89.5|92.5|85.0|82.5|77.0|83.5|76.5|74.5|85.5|86.5|
>
> *Note*: Following the recent evaluation of text-to-image benchmark [5], the human alignment score is evaluated with balanced accuracy (%) [5] across our defined four spatial domains.
>
> [1] Thinking in Space: How Multimodal Large Language Models See, Remember, and Recall Spaces. In CVPR. 2025.
>
> [2] MMSI-Bench: A Benchmark for Multi-Image Spatial Intelligence. arXiv:2505.23764. 2025.
>
> [3] Space-10: A Comprehensive Benchmark for Multimodal Large Language Models in Compositional Spatial Intelligence. arXiv:2506.07966. 2025.
>
> [4] Easier Painting Than Thinking: Can Text-to-Image Models Set the Stage, but Not Direct the Play? arXiv:2509.03516. 2025.
>
> [5] The Balanced Accuracy and Its Posterior Distribution. In ICPR. 2010.

---

> ### Author Response · Authors · 2025-11-21
> **Response to Reviewer#4 awqq (Part 2 of 3)**
>
> > **Q2: What is the design choice of including all 10 spatial constraints in every prompt? Additionally, this design poses a challenge for disentangling the contributions of the text encoder from those of the image generator, and it also complicates the generation of the corresponding QAs.**
>
> A2: These are three insightful suggestions about our core design philosophy. Here we will address each point in turn.
>
> 1. **On the motivation of involving all 10 spatial constraints in every prompt**. The decision to integrate all 10 spatial sub-domains into a single, long, information-dense prompt is deliberate and **central to our goal of moving beyond existing benchmarks.** Current benchmarks have designed valuable evaluations for foundational aspects like object presence, attributes, or other basic spatial relations. An "incremental complexity" design, while valuable, would replicate the paradigm of existing benchmarks. **Our goal is to probe how models manage multiple, often interdependent, spatial constraints simultaneously, which is a closer approximation of real-world descriptive complexity.**
> 2. **On disentangling text encoder vs. generator failure.** While our information-dense design cannot separate these two parts, we would like to clarify that our multi-dimensional design does offer powerful diagnostic clues that help approximate this disentanglement. For example:
>    - Case 1: If a model consistently fails on the Occlusion dimension across hundreds of diverse prompts, while correctly rendering other aspects like Attribute and Position within those same prompts, it strongly suggests a systemic failure in the generator's capability to render occlusion, rather than a repeated failure of the text encoder to understand the word "occluded."
>    - Case 2: If failures are random and not clustered around specific spatial concepts, it might point to more general text encoding weaknesses.
> 3. **On the Complexity of QAs Generation.** This is indeed a challenging task. This is precisely why we chose Gemini 2.5 Pro for QAs generation and adopt a human-in-the-loop refinement process (Section 2.3.2, #Lines 288-294). This ensures that despite the complexity of the generation task, the final QA pairs are accurate, targeted, and of high quality.
>
> ---
>
> > **Q3: While the intention is correct, some prompts/questions are ambiguous. How are these cases handled?**
>
> A3: Thanks for the detailed suggestion. You ask what happens if a model generates a reasonable but incorrect scenario (e.g., a rock shattering glass instead of a sound wave). We would like to clarify that **our evaluation protocol handles this by considering a generation correct only when it is faithful to our defined ground-truth answer (i.e., the original T2I prompt).**
>
> Using the example you mentioned for evaluation, the correct answer is "sound wave", not "a rock shattering glass". The T2I generation and VLM evaluation involve three cases:
> - Case 1: The T2I model generates the correct "sound wave" scene. The VLM's choice matches the ground truth, and the T2I model gets this score.
> - Case 2: The T2I model generates an incorrect "a rock shattering glass" scene, and this scene is involved in other options. The VLM will select the 'rock' option. Since this mismatches the ground truth, the model scores zero.
> - Case 3: The T2I model generates an incorrect "a rock shattering glass" scene, but this scene is not involved in other options. The VLM will select the "E: None" option. Since this mismatches the ground truth, the model scores zero.
>
> This evaluation protocol ensures that **any deviation from the prompt is wrong, regardless of how reasonable it appears.**

---

> > ### Author Response · Authors · 2025-11-21
> > **Response to Reviewer#4 awqq (Part 3 of 3)**
> >
> > > **Q4: More training details and other potential solutions will be helpful.**
> >
> > A4: Thanks for the constructive suggestion and for providing the great work of SPRIGHT. SPRIGHT offers a comprehensive study of the limitations of spatial relationships in T2I models and explores a series of training strategies and case studies related to spatial relationships. Particularly exciting is its promising data-centric path for improvement. This aligns well with our solution.
> >
> > To address your another concern, we have explored another potential method: **prompt rewriting**. We instruct Gemini 2.5 Pro to rewrite the original prompts with the specific goal of making our defined 10 spatial dimensions more explicit and unambiguous. These enhanced prompts are then sent to the text-to-image models (across diffusion-based, autoregressive-based, and unified-based) for evaluation. The prompt rewriting instruction is added in Appendix A.6.
> >
> > The results below show that prompt rewriting is a viable strategy for improving model performance. The detailed breakdown reveals three key insights:
> > 1. **Enhanced prompt decomposition is a valuable path for T2I models.** The results show that all models benefit from rewriting, confirming that a model's core ability to deconstruct complex prompts is a critical bottleneck. Notably, the improvement is more pronounced for models that initially struggle with text reasoning (e.g., SD-3.5-L shows a +2.3% gain), suggesting that improving this native capability is a promising research direction.
> > 2. **Rewriting achieves greater gains on explicit spatial relationships.** Rewriting proves highly effective at resolving textual ambiguity in categories like Position, Comparison, and Layout. This leads to substantial score increases (e.g., +4.5% in Comparison for OmniGen2, +4.7% in Position for SD-3.5-L), demonstrating its ability to clarify relational instructions.
> > 3. **Minimal impact on implicit visual reasoning.** Conversely, rewriting offers little benefit for complex visual reasoning tasks like Occlusion and Orientation. This indicates the failure stems not from text comprehension, but from a core lack of 3D and physical reasoning in the generator. Such challenges require solutions beyond prompt engineering, such as specialized fine-tuning or unified-based design for joint optimization.
> >
> > |Models|Overall|Object|Attribute|Position|Orientation|Layout|Comparison|Proximity|Occlusion|Motion|Causal|
> > |:---|:---:|:---:|:---:|:---:|:---:|:---:|:---:|:---:|:---:|:---:|:---:|
> > |SD-3.5-L   |  54.0  |  52.4  |  72.0  |  44.7  |  52.0  |  62.7  |  25.4  |  61.3  |  27.4  |  69.4  |  72.6  |
> > |+rewriting |**56.3**|**53.6**|**73.5**|**49.4**|**52.4**|**65.0**|**27.7**|**65.5**|**27.6**|**72.8**|**75.7**|
> > |OmniGen2   |  56.4  |  51.5  |  73.6  |  55.9  |  55.5  |  65.4  |  26.0  |**64.2**|  27.3  |  72.0  |  72.6  |
> > |+rewriting |**58.5**|**53.8**|**75.6**|**60.1**|**55.5**|**68.4**|**30.5**|  63.8  |**27.7**|**75.0**|**75.0**|
> > |UniWorld-V1|  54.2  |  46.8  |  71.3  |  50.1  |  53.1  |  64.0  |  26.1  |  62.0  |  26.8  |  69.6  |  72.4  |
> > |+rewriting |**55.9**|**47.6**|**72.1**|**53.8**|**53.4**|**66.0**|**29.1**|**64.8**|**26.9**|**71.5**|**73.8**|
> > |Qwen-Image |  60.6  |  61.0  |  77.2  |  55.6  |**56.7**|  69.7  |  28.6  |  67.7  |**30.8**|  78.1  |  80.2  |
> > |+rewriting |**61.7**|**62.8**|**80.8**|**57.6**|  56.4  |**70.2**|**29.7**|**68.6**|  30.4  |**79.0**|**81.1**|
> >
> > ---
> > ---
> >
> > Please let us know if any concerns remain unaddressed; we are happy to discuss them.

---

> > > ### Comment · Reviewer_awqq · 2025-11-21
> > > **Thanks for the rebuttal!**
> > >
> > > Thanks for the detailed rebuttal. The provided human studies and the alignment with VLM's is useful. I will thus increase my score and suggest authors to add this to the paper in the final version.
> > >
> > > Additionally, I still believe there is merit in incrementally increasing the difficulty of spatial prompts as that adds another dimension to the findings / analyses. The finding that prompt rewriting does not help much on complex aspects such as occlusion is also useful, please add these findings to the main paper as well.

---

> > > > ### Author Response · Authors · 2025-11-25
> > > > **Thank you for your prompt feedback and support!**
> > > >
> > > > Thank you for your prompt feedback and support. We will incorporate these modifications into the final version as you suggested.

---

### Official Review · Reviewer_WeDg · 2025-10-31

**Soundness:** 3
**Presentation:** 3
**Contribution:** 3
**Rating:** 2
**Confidence:** 5

**Summary:**

The paper introduces a new benchmark for critically evaluating the spatial understanding of current text-to-image models.

**Strengths:**

* Well-motivated problem statement
* Diverse evaluation dataset
* Well-guiding principles behind dataset construction
* Diverse results from many different models

**Weaknesses:**

* Unclarity in writing
* Lack of QA in MLLM-synthesized prompts
* Emphasis on using 77 tokens (L259) despite the fact that modern models go well beyond this context length

**Questions:**

* L85 - L86: It's not clear what the sentence means. Does it mean high-scoring samples from T2I models?  Also, which prompts are being referred to here?
* Figure 1: Are the bottom formats from different eval frameworks? If so, which ones?
* Could the authors provide comparisons to Nano Banana [1] in Figure 2 and in general?
* Could the authors provide some broader trends from benchmark results? For example, do models with LLMs as text encoders (QwenImage [2], for example) show better performance than the ones without?
* "Refuse to answer (not guess)" -- how is this taken into consideration during the evals?
* Could the authors consider using the SPRIGHT [3] fine-tuning strategy and see if that helps? The strategy is to only train on images where the number of objects exceeds a certain threshold.
* The SFT experiments include no diffusion-based models. Could the authors include an experiment with a diffusion-based model, say QwenImage?

Misc:

1. Some missing references: SPRIGHT [1], HEIM [4]

References:

[1] Nano Banana; https://gemini.google/us/overview/image-generation/?hl=en

[2] Qwen-Image Technical Report; Wu et al.; 2025

[3] Getting it Right: Improving Spatial Consistency in Text-to-Image Models; Chatterjee et al.; 2024

[4] Holistic Evaluation of Text-To-Image Models; Lee et al.; 2023

---

> ### Author Response · Authors · 2025-11-21
> **Response to Reviewer#3 WeDg (Part 1 of 2)**
>
> We sincerely thank Reviewer `WeDg` for the time and detailed feedback. We appreciate your positive comments on our "well-motivated problem", "guiding principles" and "diverse results." In response to your suggestions, we have enriched our work with new experiments, additional comparisons, and a more thorough discussion of trends. We hope the current manuscript now merits your reconsideration for acceptance.
>
> > **Q1: Unclear writing in Line 85-86.**
>
> A1: Specifically, as stated in \#Line 85-91 and \#Line 470-481, we construct *another* dataset (SpatialT2I) for supervised fine-tuning. More importantly, SpatialT2I is constructed separately and has no overlap with our evaluation benchmark. The construction of SpatialT2I involves two stages:
> 1. **Stage 1: Prompt and Omni-dimensional QAs generation**. This stage follows the same principles of our SpatialGenEval in Section 2.3. Totally, we obtain another 1,230 prompts and 12,300 QAs.
> 2. **Stage 2: Rewrite prompt to obtain text-image pair**. We curate image outputs from all 14 T2I models with average scores above 50% (Table 2, 9 in the manuscript), along with their generation prompts. These are processed by a strong MLLM (e.g., Gemini 2.5 Pro, \#Line 1143-1170) to produce mildly rewritten prompts that better match the corresponding images, improving text-image consistency while preserving information density and all dimensions of spatial intelligence. Here we omit the "Design" scenes (130 prompts) due to low image quality. Totally, we obtain 15,400 image-text pairs, i.e., (1230-130)\*14=15400.
>
> ---
>
> > **Q2: Figure 1: Are the bottom formats from different eval frameworks? If so, which ones?**
>
> A2: Yes. The bottom-left image sources from the recent TIIF-Bench [1], while the bottom-right image is from our SpatialGenEval. We have updated the reference in the caption of Figure 1.
>
> [1] TIIF-Bench: How Does Your T2I Model Follow Your Instructions?. arXiv preprint arXiv:2506.02161. 2025
>
> ---
>
> > **Q3: Could the authors provide comparisons to Nano Banana in Figure 2 and in general?**
>
> A3: Thanks for pointing out this most recent generative model (Nano Banana, Gemini-2.5-Flash-Image). We have updated Nano Banana in Figure 2 and Table 2. Nano Banana achieves the best performance (61.7%) but still reaches the 60-point passing threshold, highlighting that SpatialGenEval remains suitable and complex enough to challenge current SOTA generative models.
>
> The results of Nano Banana (based on the evaluation of GPT-4o and Qwen2.5-VL-72B) are shown as follows.
>
> |Models|VLMs|Overall|Object|Attribute|Position|Orientation|Layout|Comparison|Proximity|Occlusion|Motion|Causal|
> |:---|:---:|:---:|:----:|:---:|:---:|:---:|:---:|:---:|:---:|:---:|:---:|:---:|
> |Nano Banana|GPT-4o|61.6|68.0|74.5|56.3|62.0|69.0|26.4|70.7|36.1|74.6|78.3|
> |Nano Banana|Qwen2.5-VL-72B|61.7|58.5|75.3|55.5|58.9|70.9|31.8|68.7|33.5|81.4|82.2|
>
> ---
>
> > **Q4: Whether the authors can provide broader trends like do models with LLMs as text encoders show better performance than the ones without?**
>
> A4: Indeed, models with stronger text encoders, particularly those leveraging powerful LLMs, consistently outperform those with standard CLIP encoders. For example, Qwen-Image (60.6%), which uses a powerful LLM encoder, is the top-performing open-source model. Similarly, the FLUX.1 models (56.5-58.5%) and SD-3 models (54.0-54.6%), which use improved text encoders (like T5) alongside CLIP, significantly outperform older models like SD-1.5 (28.5%). This strongly suggests that a deeper understanding of complex, information-dense prompts is a critical component of achieving spatial intelligence. We have added this analysis to Section 3.2.

---

> > ### Author Response · Authors · 2025-11-21
> > **Response to Reviewer#3 WeDg (Part 2 of 2)**
> >
> > > **Q5: "Refuse to answer (not guess)" -- how is this taken into consideration during the evals?**
> >
> > A5: During the evaluation stage, the ground-truth answer is always one of options "A-D", while another option "E: None" is designed to avoid cases where the model is forced to select incorrect answers where the required visual element is missing. This mechanism ensures we are measuring successful generation, not forced-choice guessing.
> >
> > To demonstrate this process, we use an extreme example as follows. We provide the model with only the question, without the corresponding image. As expected, the model selects option "E: None" due to the missing visual input.
> >
> >     Example (No Image Input, based on Qwen2.5-VL-72B, Scene: Forest, ID: 000001, Accuracy: 0/10):
> >     [
> >     "E: None - The image does not provide clear depiction or labels for precise object categories such as \"apple\" vs \"orange\" or \"squirrels\" vs \"chipmunks,\" nor the presence of a \"fern.\"",
> >     "E: None - The image does not give clear details about the color or material of the lantern.",
> >     "E: None - There's no clear path indicated in the image which could guide the location of the boy relative to it.",
> >     "E: None - The image lacks clear directional indicators regarding the facing orientations of the individuals.",
> >     "E: None - The arrangement of the squirrels cannot be determined precisely from the image.",
> >     "E: None - The image does not offer a definitive indication of the height difference between the boy and the girl.",
> >     "E: None - The placement of the blue lantern cannot be determined accurately from the image.",
> >     "E: None - The visibility status of the boy cannot be confirmed based solely on the given image.",
> >     "E: None - The image does not show any evident motion interactions between the girl and the boy.",
> >     "E: None - The image doesn't provide information on the illumination sources affecting the faces of the boy and girl."
> >     ]
> >
> > ---
> >
> > > **Q6: Could the authors consider using SPRIGHT's fine-tuning strategy and see if that helps? The strategy is to only train on images where the number of objects exceeds a certain threshold.**
> >
> > A6: Thanks for the insightful suggestion and for providing the great work of SPRIGHT. SPRIGHT surprisingly found that training on just 444 images with a large number of subjects yields substantial improvements in spatial consistency. **We are also surprised that its focus on a large number of subjects aligns well with our concept of an information-dense setting.**
> >
> > However, when attempting to reproduce this, we found that only 224 of 444 images (from Segment Anything) have been released. The remaining 220 images (from LAION) are inaccessible since their parent images are not public. This issue is confirmed by the authors themselves in a public discussion (https://github.com/SPRIGHT-T2I/SPRIGHT/issues/6).
> >
> > Based on this, we **fine-tune OmniGen2 for 2/4/6/8 epochs using the available 224 images**. The fine-tuned results (overall: 56.5/56.4/56.6/56.3) remain similar to the original OmniGen2 (overall: 56.4). The possible reason maybe the small number of images. Despite this result, we still believe that **leveraging information-dense settings by increasing object number is a promising approach to improve fine-grained perception and generation**, and this remains a valuable area for future work.
> >
> > ---
> >
> > > **Q7: The SFT experiments include no diffusion-based models. Could the authors include an experiment with a diffusion-based model?**
> >
> > A7: Following your suggestion, we have conducted a diffusion-based model (Stable Diffusion XL) based on our SpatialT2I dataset. The results below show that our SpatialT2I achieves consistent improvements (overall +4.2%). We have updated these results in Section 4 and Table 5.
> >
> > |Models|Overall|Object|Attribute|Position|Orientation|Layout|Comparison|Proximity|Occlusion|Motion|Causal|
> > |:---|:---:|:---:|:---:|:---:|:---:|:---:|:---:|:---:|:---:|:---:|:---:|
> > |SD-XL      |41.2|25.7|52.8|32.0|40.9|49.3|19.1|50.7|22.4|56.7|62.0|
> > |+SpatialT2I|45.4|29.0|55.7|38.6|48.3|53.6|23.0|54.8|27.2|61.6|63.2|
> >
> > ---
> >
> > > **Q8: Some missing references.**
> >
> > A8: Thanks for pointing out these omissions. We have added these references (SPRIGHT [2] and Holistic Evaluation of Text-To-Image Models [3]) to our related work section (Section 4) and future work section (Section A.9.4). All relevant works are appropriately cited in the revised manuscript.
> >
> > [2] Getting It Right: Improving Spatial Consistency In Text-to-Image Models. In ECCV. 2024.
> >
> > [3] Holistic Evaluation of Text-to-Image Models. In NeurIPS. 2023.
> >
> > ---
> >
> > Please let us know if any concerns remain unaddressed; we are happy to discuss them.

---

> > > ### Comment · Reviewer_WeDg · 2025-11-21
> > >
> > > > A1:
> > >
> > > Could this be made a part of the main text more explicitly?
> > >
> > > > A3:
> > >
> > > Could the authors include these results in the main text along with one other closed model such as Seed Dream 4.0 or GPT-Image? Also, a discussion on how a close VLM (GPT-4o) and an open VLM (Qwen2.5-VL-72B) yield similar trends in the numbers could be worthwhile to include.
> > >
> > >
> > > > To demonstrate this process, we use an extreme example as follows. We provide the model with only the question, without the corresponding image. As expected, the model selects option "E: None" due to the missing visual input.
> > >
> > > Interesting experiment! However, I think the model should have rejected outputting further the moment it discovered there was no image input at all. Could the authors provide any comment on that?
> > >
> > > > Based on this, we fine-tune OmniGen2 for 2/4/6/8 epochs using the available 224 images. The fine-tuned results (overall: 56.5/56.4/56.6/56.3) remain similar to the original OmniGen2 (overall: 56.4). The possible reason maybe the small number of images. Despite this result, we still believe that leveraging information-dense settings by increasing object number is a promising approach to improve fine-grained perception and generation, and this remains a valuable area for future work.
> > >
> > > Sorry for being unclear. I meant if the authors could follow the SPRIGHT strategy of fine-tuning, i.e., taking a small number of images from the SpatialGenEval training set and fine-tuning a model on that subset to see if the performance gets any better. Additionally, showing that for both diffusion and non-diffusion models would be quite advantageous.
> > >
> > > Based on the discussions further, I will consider raising my score.

---

> ### Author Response · Authors · 2025-11-25
> **Follow up Response to Reviewer#3 WeDg**
>
> > **Q1: Could this be made a part of the main text more explicitly?**
>
> A1: Certainly. We have updated these statements in the introduction (Line 85-90) and Section 4 (Line 470-481). Hope the current version meets your expectations.
>
> ---
>
> > **Q2: Could the authors include these results in the main text along with one other closed model, such as Seed Dream 4.0 or GPT-Image? Also, a discussion on how a closed VLM (GPT-4o) and an open VLM (Qwen2.5-VL-72B) yield similar trends in the numbers could be worthwhile to include.**
>
> A2: Firstly, in the revised version, the evaluated closed-source T2I models include `DALL-E-3`, `GPT-Image-1`, `Nano Banana`, and `Seed Dream 4.0`. Among them, Seed Dream 4.0 achieves the best performance (i.e., 62.7% based on Qwen2.5-VL-72B). Moreover, the closed-source T2I models show a higher performance gap in the most challenging Spatial Reasoning and Interaction compared with open-source models. We have updated these in Table 2/10, Figure 2, and discussed them in Section 3.2.
>
> Secondly, both closed-source VLM (GPT-4o) and open-source VLM (Qwen2.5-VL-7B) yield similar model rankings and numbers, except for the specific Object sub-domain. The probable reason appears to be the complexity of our defined questions in this sub-domain, for example:
>
>     Example 1: Which option best describes the object categories explicitly mentioned in the scene?
>     Options: A: Bear, cubs, salmon, river, pine trees, B: Bear, salmon, river, fish, stones, C: Wolf, cubs, salmon, river, trees, D: Bear, rocks, salmon, lake, grass, E: None
>     Example 2: Besides people and a vehicle, what other man-made and natural features are mentioned in the scene？
>     Options: A: A bridge and a river, B: A road and a forest, C: A stone wall and a valley, D: A house and a lake, E: None
>
> However, **the overall consistency of the ranking and number validates the robustness of our benchmark and the selected evaluator**. This discussion has been added to Section 3.2 (Line 419-424).
>
> ---
>
> > **Q3: I think the model should have rejected outputting further the moment it discovered there was no image input at all. Could the authors provide any comment on that?**
>
> A3: Firstly, we would like to clarify that **this extreme example is only to demonstrate the model's output in cases where the prompt is unfaithful to the image. During our implementation, each evaluation involves a generated image and the corresponding questions.**
>
> Secondly, in this extreme example (no image input), VLM also outputs and selects "E: None" because of our meta-instruction defined in Appendix A.4 (Line 1121-1122). **We explicitly instruct the VLM evaluator to select the best-matching option (e.g., one of the options from A/B/C/D/E). This instruction indicates that the model must select one of the options and provide its reasoning.** In this case, when none of image as input, VLM can only select the ``E: None'' option as expected.
>
> Finally, in a real evaluation (an image along with its questions), it typically involves the following two cases (ignoring VLM prediction errors):
> - Case 1: The T2I model generates the correct scene, and the VLM selects the ground-truth option.
> - Case 2: The T2I model generates an incorrect scene. The VLM then selects an incorrect option (A/B/C/D) based on the actually generated image, or selects "E: None" when none of the options match.
>
> ---
>
> > **Q4: I meant if the authors could follow the SPRIGHT strategy of fine-tuning, i.e., taking a small number of images from the SpatialGenEval training set and fine-tuning a model on that subset to see if the performance gets any better. Additionally, showing that for both diffusion and non-diffusion models would be quite advantageous.**
>
> A4: Thank you for the follow-up. To address your concern, we have conducted two additional experiments as follows.
>
> **Ablation study of SpatialT2I and data scaling trend.** To assess the impact of data quality and quantity in SpatialT2I, we conduct two experiments as follows.
>
> 1. Firstly, we follow SPRIGHT and select three subsets arranged by increasing performance in Table 2, i.e., Unipic-v2 (54.3), Bagel (57.0), and Qwen-Image (60.6). Each subset includes 1100 text-image pairs. As shown in Figure 7, fine-tuning on both the diffusion model (SD-XL) and non-diffusion model (OmniGen2) reveals that all subsets yield performance gains, and higher-scoring ones contribute more significantly.
>
> 2. Secondly, we observe a data scaling trend in Figure 7, as performance consistently improves when increasing training data from 0% to 100%, by progressively adding higher-scoring subsets.
>
> These findings reveal the value of exploring information-dense, spatial-aware data and suggest that further scaling is a promising avenue for future work.
>
> ---
>
> Thanks again for your valuable feedback and for helping us improve the quality of our paper! Please let us know if any concerns remain unaddressed; we are happy to discuss them.

---

> > ### Comment · Reviewer_WeDg · 2025-11-25
> >
> > Nice finding, especially on the data scaling trends. I have raised my score to honor the thoroughness of the rebuttal.

---

> > > ### Author Response · Authors · 2025-11-25
> > > **Thanks for your prompt feedback and support!**
> > >
> > > Thanks for taking the time to review our rebuttal. We are glad that our response has addressed your concerns. Once again, we appreciate your detailed and helpful review, as well as your prompt and positive feedback!

---

### Official Review · Reviewer_WonS · 2025-10-31

**Soundness:** 2
**Presentation:** 3
**Contribution:** 2
**Rating:** 4
**Confidence:** 4

**Summary:**

The paper proposes a new T2I benchmark, SpatialGenEval, designed to evaluate the spatial understanding of T2I models. This benchmark features long, information-dense prompts that incorporate complex spatial descriptions, offering a more challenging evaluation compared to previous benchmarks. It categorizes the prompts into four main domains and ten subdomains to enable a more detailed analysis of model performance.
The paper also introduces a new evaluation metric that employs a Vision-Language Model (VLM) to analyze the generated images and answer multiple-choice questions based on them. Using this benchmark, the authors conduct a comprehensive evaluation of four different architecture of T2I models, covering both pioneering and state-of-the-art approaches. The results highlight the challenges these models face when generating images with more advanced spatial relations—particularly those involving 3D spatial reasoning.
Finally, the paper demonstrates the utility of the proposed benchmark by fine-tuning a T2I model using the benchmark prompts. The fine-tuned model shows some improvements in spatial understanding in image generation.

**Strengths:**

- The paper introduces a new benchmark with complex and information-dense prompts, which significantly improve upon previous benchmarks that rely on shorter prompts and limited number of spatial relations.
- The paper also introduces a new evaluation protocol that incorporates a vision-language model (VLM) to assess the completeness of generated images through visual question answering, specifically designed to evaluate spatial understanding.
- The benchmark includes human experts in refining prompts and evaluating questions to ensure quality, soundness, and appropriate complexity.
- The benchmark reveals performance gaps across different categories of spatial relations, enabled by the detailed annotations proposed in the benchmark.
- The paper illustrates the usefulness of the evaluation framework for improving model performance in text-to-image generation.
- Well-document and easy to follow paper. Especially figures make the paper easy to follow and understand.

**Weaknesses:**

• Although the paper proposes a benchmark with complex spatial relations, the evaluation protocol raises some concerns. As demonstrated by several prior works VLMs still struggle with spatial understanding—particularly in 3D spatial relations[1, 2, 3]. Therefore, the interpretation of the results is limited, as it remains unclear whether the observed issues stem from the T2I model itself or from the evaluation protocol.

• Although the authors demonstrate that fine-tuning improves performance, the paper does not explore other potential methods or strategies to address the lack of spatial understanding in T2I models.

• While the authors incorporate a comprehensive human-in-the-loop process for dataset creation, the study does not include a comparison against a human baseline. This is significant, especially when introducing a new evaluation protocol, as it is unclear whether the proposed metric aligns with human judgment. The significant of VQA settings for evaluation is drop without showing this can compared with previous methods.

• The error analysis is based solely on four domain types and relies on numerical accuracy. A qualitative, human-centered analysis of failure cases is needed to provide deeper insights into model weaknesses.

[1] Liu et al. Visual Spatial Reasoning, TACL 2023.

[2] Chen et al. SpatialVLM: Endowing Vision-Language Models with Spatial Reasoning Capabilities, CVPR 2024.

[3] Zhang et al. Do Vision-Language Models Represent Space and How? Evaluating Spatial Frame of Reference under Ambiguities, ICLR 2025.

**Questions:**

There are no additional questions beyond those asked in weaknesses.

---

> ### Author Response · Authors · 2025-11-21
> **Response to Reviewer#2 WonS (Part 1 of 3)**
>
> We sincerely thank Reviewer `WonS` for the thorough summary and constructive feedback. We are encouraged that you recognized our benchmark as a "significant improvement" over previous work and found the paper to be "well-document and easy to follow". Below, we provide our point-by-point responses and hope to address your concerns.
>
> > **Q1: Concern about the reliance of VLM for evaluation.**
>
> A1: We acknowledge the known limitations of VLMs in handling highly complex spatial reasoning (particularly in 3D spatial relations), just like the provided references. However, we would like to clarify that current leading VLMs are **suitable** and **capable** for evaluating the questions in our SpatialGenEval benchmark, based on three key aspects:
> 1. **Design philosophy: The primary burden of spatial intelligence of our SpatialGenEval lies in the generative T2I model, not the VLM evaluator.** The T2I model must integrate the long, information-dense prompt to create an image with all defined spatial relationships. In contrast, the VLM's task is simplified to simple visual checks based on the direct visual evidence. **This design choice decouples the complex reasoning required for generation from the simpler verification required for evaluation.**
> 2. **Problem difficulty: Unlike benchmarks designed to push the boundaries of VLM spatial understanding, SpatialGenEval focuses on evaluating generative task and consists of simpler QAs.** Unlike recent spatial intelligence benchmarks such as VSI-Bench [1], MMSI-Bench [2], and Space-10 [3], which target *understanding* task and focus on more abstract or complex long-range reasoning (e.g., estimating real-world distances or path planning), SpatialGenEval is targeted to *generative* task and concentrates on evaluating more simpler visual relationships like relative position, layout, occlusion, and interaction. These tasks can be verified directly from the image's visual content, making them more suitable for current VLMs.
> 3. **Human alignment: Current leading VLMs align well with human evaluation in our SpatialGenEval.** Our new human alignment study provides direct numerical validation that current VLMs are capable of handling our designed questions. The results reveal an exceptionally high correlation between the VLM and human evaluations, i.e., Qwen2.5-VL-72B (80.4%) and GPT-4o (78.8%). The detailed settings and results of our human alignment study please refer to the following **Q3**. We have added the human alignment study to Section 3.2.
>
> [1] Thinking in Space: How Multimodal Large Language Models See, Remember, and Recall Spaces. In CVPR. 2025.
>
> [2] MMSI-Bench: A Benchmark for Multi-Image Spatial Intelligence. arXiv:2505.23764. 2025.
>
> [3] Space-10: A Comprehensive Benchmark for Multimodal Large Language Models in Compositional Spatial Intelligence. arXiv:2506.07966. 2025.

---

> > ### Author Response · Authors · 2025-11-21
> > **Response to Reviewer#2 WonS (Part 2 of 3)**
> >
> > > **Q2: More exploration of other potential methods or strategies beyond fine-tuning.**
> >
> > A2: We agree that our work primarily focuses on constructing the benchmark and using SFT as a "proof-of-concept" to demonstrate the utility of our data. Our attempt is not to provide an exhaustive survey of all possible improvement methods, but rather to validate a *data-centric* approach as one viable path forward.
> >
> > To address your concern, we have explored another potential method: **prompt rewriting**. We instruct Gemini 2.5 Pro to rewrite the original prompts with the specific goal of making our defined 10 spatial dimensions more explicit and unambiguous. These enhanced prompts are then sent to the text-to-image models (across diffusion-based, autoregressive-based, and unified-based) for evaluation. The prompt rewriting instruction is added in Appendix A.6.
> >
> > The results below show that prompt rewriting is a viable strategy for improving model performance. The detailed breakdown reveals three key insights:
> > 1. **Enhanced prompt decomposition is a valuable path for T2I models.** The results show that all models benefit from rewriting, confirming that a model's core ability to deconstruct complex prompts is a critical bottleneck. Notably, the improvement is more pronounced for models that initially struggle with text reasoning (e.g., SD-3.5-L shows a +2.3% gain), suggesting that improving this native capability is a promising research direction.
> > 2. **Rewriting achieves greater gains on explicit spatial relationships.** Rewriting proves highly effective at resolving textual ambiguity in categories like Position, Comparison, and Layout. This leads to substantial score increases (e.g., +4.5 in Comparison for OmniGen2, +4.7 in Position for SD-3.5-L), demonstrating its ability to clarify relational instructions.
> > 3. **Minimal impact on implicit visual reasoning.** Conversely, rewriting offers little benefit for complex visual reasoning tasks like Occlusion and Orientation. This indicates the failure stems not from text comprehension, but from a core lack of 3D and physical reasoning in the generator. Such challenges require solutions beyond prompt engineering, such as specialized fine-tuning or unified-based design for joint optimization.
> >
> > |Models|Overall|Object|Attribute|Position|Orientation|Layout|Comparison|Proximity|Occlusion|Motion|Causal|
> > |:---|:---:|:---:|:---:|:---:|:---:|:---:|:---:|:---:|:---:|:---:|:---:|
> > |SD-3.5-L   |  54.0  |  52.4  |  72.0  |  44.7  |  52.0  |  62.7  |  25.4  |  61.3  |  27.4  |  69.4  |  72.6  |
> > |+rewriting |**56.3**|**53.6**|**73.5**|**49.4**|**52.4**|**65.0**|**27.7**|**65.5**|**27.6**|**72.8**|**75.7**|
> > |OmniGen2   |  56.4  |  51.5  |  73.6  |  55.9  |  55.5  |  65.4  |  26.0  |**64.2**|  27.3  |  72.0  |  72.6  |
> > |+rewriting |**58.5**|**53.8**|**75.6**|**60.1**|**55.5**|**68.4**|**30.5**|  63.8  |**27.7**|**75.0**|**75.0**|
> > |UniWorld-V1|  54.2  |  46.8  |  71.3  |  50.1  |  53.1  |  64.0  |  26.1  |  62.0  |  26.8  |  69.6  |  72.4  |
> > |+rewriting |**55.9**|**47.6**|**72.1**|**53.8**|**53.4**|**66.0**|**29.1**|**64.8**|**26.9**|**71.5**|**73.8**|
> > |Qwen-Image |  60.6  |  61.0  |  77.2  |  55.6  |**56.7**|  69.7  |  28.6  |  67.7  |**30.8**|  78.1  |  80.2  |
> > |+rewriting |**61.7**|**62.8**|**80.8**|**57.6**|  56.4  |**70.2**|**29.7**|**68.6**|  30.4  |**79.0**|**81.1**|

---

> ### Author Response · Authors · 2025-11-21
> **Response to Reviewer#2 WonS (Part 3 of 3)**
>
> > **Q3: Lack of comparison of human alignment study.**
>
> A3: Thanks for your insightful suggestion. Following your suggestion, we conduct the **Human Alignment Study across different MLLMs and 10 sub-domains based on the Qwen-Image model**.
> - **Settings of human alignment study**. Following our VLM evaluation setup (\#Line 315-323), five human annotators work independently to *select the best option* based solely on the *given image* and *10 questions*, with no access to the original text-to-image prompt to prevent "leakage". We randomly select 8 images from each of the 25 scenes, resulting in a total of 200 images. We follow the recent T2I benchmark [4] and adopt the *balanced accuracy* [5] as the overall human alignment score for alignment evaluations. All human annotators are experts in text-to-image generation, ensuring the quality and validity of the annotations.
> - **Results of human alignment study**. The human alignment scores across 10 sub-domains are presented in the table below. We find that: (1) Overall alignment: The three selected models all achieve high scores. Gemini 2.5 Pro ranks the highest, followed by Qwen2.5-VL-72B and GPT-4o. (2) Alignment across sub-domains: We find that the human alignment correlates with the difficulty of each sub-domain. **The alignment is higher on simpler dimensions like Spatial Foundation/Perception/Interaction, while lower on the Spatial Reasoning sub-domain. Despite this, the alignment score still nears 80%**, validating the effectiveness of our chosen open-source and closed-source models as evaluators in SpatialGenEval.
>
> | VLMs         |Overall|Object|Attribute|Position|Orientation| Layout | Comparison | Proximity | Occlusion | Motion | Causal |
> |:-------------|:-----:|:----:|:-------:|:------:|:---------:| :----: | :--------: | :-------: | :-------: | :----: | :----: |
> |Qwen2.5-VL-72B|  80.4 | 84.5 |  89.5   |  82.5  |   77.5    |  69.5  |    80.0    |    71.0   |    69.0   |  84.0  |  85.5  |
> |GPT-4o        |  78.8 | 80.5 |  85.0   |  81.0  |   78.5    |  70.5  |    79.5    |    72.0   |    66.0   |  80.5  |  86.0  |
> |Gemini 2.5 Pro|  84.2 | 89.5 |  92.5   |  85.0  |   82.5    |  77.0  |    83.5    |    76.5   |    74.5   |  85.5  |  86.5  |
>
> *Note*: Following the recent evaluation of text-to-image benchmark [4], the human alignment score is evaluated with balanced accuracy (%) [5] across our defined four spatial domains.
>
> [4] Easier Painting Than Thinking: Can Text-to-Image Models Set the Stage, but Not Direct the Play? arXiv:2509.03516. 2025.
>
> [5] The Balanced Accuracy and Its Posterior Distribution. In ICPR. 2010.
>
> ---
>
> > **Q4: Error analysis is solely numerical; a qualitative, human-centered analysis is needed.**
>
> A4: Thanks for the constructive suggestion. We have analyzed the error cases around spatial perception, reasoning, and interaction based on current top-performing models like Qwen-Image, GPT-Image-1, and Bagel. Additionally, another data example is also shown in the Appendix (Page 27). In the future, we will release the website to involve more data samples for a clear visualization.
>
> ---
>
> Please let us know if any concerns remain unaddressed; we are happy to discuss them.

---

> > ### Comment · Reviewer_WonS · 2025-11-25
> >
> > Thank you for the additional experiments and discussion. The paper has been updated to reflect these changes. I am satisfied with the human alignment to confirm the evaluator’s performance. I raised my score because the response addressed my concerns.

---

> > > ### Author Response · Authors · 2025-11-26
> > > **Thanks for your prompt feedback and support!**
> > >
> > > We are grateful to hear our feedback has addressed your concerns. We sincerely appreciate your recognition of our contributions and the increased score. Thank you once again for your time and effort in reviewing our paper!

---

### Official Review · Reviewer_vwE4 · 2025-11-06

**Soundness:** 2
**Presentation:** 4
**Contribution:** 3
**Rating:** 8
**Confidence:** 4

**Summary:**

The paper introduces a new text-to-image generation benchmark focused on testing spatial capabilities. The benchmark consists of 1.2k prompts, generated by an LLM prompted with different combinations of domains and themes, and covering 10 axes of spatial capability. After refinement, for evaluation, an MLLM is given the prompts and asked to produce QA pairs aligning with the 10 spatial aspects. Newer T2I models are tested on this benchmark and findings show that spatial comparison and occlusion are two main bottlenecks in improving spatial capabilities.

**Strengths:**

The work introduces previously untested axes of T2I capability, and provides a large benchmark designed to test models along these new dimensions. The benchmark consists of LLM-generated prompts with automated model-based evaluation, where the generation prompts and evaluation QA pairs are refined manually by humans. While the integrity of the model-based evals could be contested, the diversity of prompts and structured scoring (evenly across the clearly defined 10 sub-domains) is clear.

The benchmark also remains inclusive through information-dense prompts that are short enough to fairly evaluate older T2I models while being difficult enough to distinguish between frontier models.

**Weaknesses:**

In the analysis of error rates and failure cases, the authors mention "This clear hierarchy demonstrates that models learn skills in a specific order". However, the hierarchy is not clear from Figure 5; it seems for example that motion, which appears after relational reasoning, is already easier. Some clarification would be helpful here.

It is unclear how impactful the contribution of the training data is. While the results show that SFT on the new data improves performance on their benchmark, this is to be expected since the method of training data generation/collection is closely tied to the benchmark. It would be helpful to see numbers from other benchmarks showing whether this improvement also carries into other metrics and maintains the same rank correlation.

**Questions:**

1. What informs the choice of Gemini 2.5 Pro as the model to create T2I prompts?
2. How strongly do individual per-question scores in the QA portion align with human judgment? While the strong correlation with other benchmark numbers is convincing, if the goal is a breakdown of model capabilities by sub-domain, it is important that these scores also correlate strongly with human or other benchmark judgment.
3. In human refinement of the QA pairs, you mention revising questions that can be answered from the prompt alone. Is this determined by whether a human can answer correctly without an image, or a model? Often, models can still end up being biased towards the correct answer, even if it is not apparent from human inspection.
4. Is there any analysis of bias towards/against image quality or image style, independent of spatial correctness? E.g., do models that output photorealistic vs digital art-style images exhibit higher or lower scores on the benchmark despite having similar true spatial capabilites?

---

> ### Author Response · Authors · 2025-11-21
> **Response to Reviewer#1 vwE4 (Part 1 of 2)**
>
> We sincerely thank reviewer `vwE4` for the effort and constructive feedback on our manuscript. We are very happy that you find our work to be well-motivated, clear, and well-presented. Below, we provide our point-by-point response and hope to address your concerns.
>
> ---
>
> > **Q1: What informs the choice of Gemini 2.5 Pro to create T2I prompts?**
>
> A1: Thanks for your detailed suggestion. **The choice of Gemini 2.5 Pro stems from its strong creative ability (less repetition than others across the prompts within the same scene)**. Specifically, the construction of SpatialGenEval benchmark concentrates on two abilities as follows.
> 1. **Strong instruction following ability**: As stated in Section 2.3.1 (\#Line 261-264), the prompt generation process must strictly follow the instructions to generate each text-to-image prompt based on a given scene (e.g., classroom) while seamlessly integrating all 10 pre-defined spatial sub-domains.
> 2. **Strong creative ability (less repetition)**: As stated in Appendix A.4 (\#Line 1003-1004), we instruct the model to generate 50 distinct text-to-image prompts all at once. This requires a strong creative ability to avoid generating repetitive outputs.
>
> During our past implementation, we have conducted a controlled experiment where we instructed three top-performing MLLMs (Gemini 2.5 Pro, GPT-4o, and Qwen2.5-VL-72B) for prompt generation. We found that all three models demonstrate strong instruction following ability, but **Gemini 2.5 Pro outperformed the others in terms of creative ability**. This observation stems from (1) consistent decision from all five human annotators, and (2) Gemini 2.5 Pro shows *lower sentence similarity* [1] across all text-to-image prompts within the same scene, i.e., **Gemini 2.5 Pro (0.4938) < GPT-4o (0.5125) < Qwen2.5-VL-72B (0.5548)**. We have added this discussion to Appendix A.9.1.
>
> [1] Sentence-BERT: Sentence embeddings using siamese BERT-Networks. In EMNLP. 2019.
>
> ---
>
> > **Q2: While the strong correlation with other benchmark numbers is convincing. How strongly do the VLM scores (across 10 sub-domains) align with human judgment?**
>
> A2: Thanks for your insightful suggestion. Following your suggestion, we conduct the **Human Alignment Study across different MLLMs and 10 sub-domains based on the Qwen-Image model**, and add these results to Section 3.2 (\#Line 425-463).
>
> - **Settings of human alignment study**. Following our VLM evaluation setup (\#Line 316-320), five human annotators work independently to **select the best option based solely on the given image and 10 questions**, with no access to the original text-to-image prompt to prevent "leakage". We randomly select 8 images from each of the 25 scenes, resulting in a total of 200 images. We follow the recent T2I benchmark [2] and adopt the *average balanced accuracy* [3] as the overall human alignment score for alignment evaluations. All human annotators are experts in text-to-image generation, ensuring the quality and validity of the annotations.
> - **Results of human alignment study**. The human alignment scores across 10 sub-domains are presented in the table below. We find that: (1) Overall alignment: The three selected models all achieve high scores. Gemini 2.5 Pro ranks the highest, followed by Qwen2.5-VL-72B and GPT-4o. (2) Alignment across sub-domains: We find that the human alignment correlates with the difficulty of each sub-domain. **The alignment is highest on simpler dimensions like Spatial Foundation and Perception, and lowest on the Spatial Reasoning sub-domain. Despite this, the alignment score still nears 80%**, validating the effectiveness of our chosen open-source and closed-source models as evaluators in SpatialGenEval.
>
> | VLMs         |Overall|Object|Attribute|Position|Orientation| Layout | Comparison | Proximity | Occlusion | Motion | Causal |
> |:-------------|:-----:|:----:|:-------:|:------:|:---------:| :----: | :--------: | :-------: | :-------: | :----: | :----: |
> |Qwen2.5-VL-72B|  80.4 | 84.5 |  89.5   |  82.5  |   77.5    |  69.5  |    80.0    |    71.0   |    69.0   |  84.0  |  85.5  |
> |GPT-4o        |  78.8 | 80.5 |  85.0   |  81.0  |   78.5    |  70.5  |    79.5    |    72.0   |    66.0   |  80.5  |  86.0  |
> |Gemini 2.5 Pro|  84.2 | 89.5 |  92.5   |  85.0  |   82.5    |  77.0  |    83.5    |    76.5   |    74.5   |  85.5  |  86.5  |
>
> *Note 1*: Following the recent evaluation of the text-to-image benchmark [2], the human alignment score is evaluated with average balanced accuracy (%) [3] across our defined four spatial domains.
>
> *Note 2*: Spatial Foundation (Object, Attribute), Spatial Perception (Position, Orientation, Layout), Spatial Reasoning (Comparison, Proximity, Occlusion), Spatial Interaction (Motion, Causal).
>
> [2] Easier Painting Than Thinking: Can Text-to-Image Models Set the Stage, but Not Direct the Play?. arXiv:2509.03516. 2025.
>
> [3] The Balanced Accuracy and Its Posterior Distribution. In ICPR. 2010.

---

> ### Author Response · Authors · 2025-11-21
> **Response to Reviewer#1 vwE4 (Part 2 of 2)**
>
> > **Q3: Clarification on the methodology for filtering 'prompt-only answerable' QAs. Specifically, is this filtering based on human or model evaluation? Also, how does the process mitigate potential model biases, such as the model's tendency to favor the correct answer even with human refinement.**
>
> A3: Firstly, we would like to clarify that **this process is implemented directly by human annotators (Section 2.3.2)**. Specifically, the explicit task of human annotators is to **identify and eliminate the question containing an explicit answer**, where the answer lies in the question. For example, a question we would be revised as follows.
>
>     - Question: What is the layout of the leaves that are arranged in a circle?
>     - Revise: The answer ("in a circle") is given in the question itself. Human annotators need to rephrase such questions to enforce dependency on visual evidence, i.e., changing it to "What is the layout of the leaves in the image?".
>
> Secondly, to mitigate the potential biases from model predictions (e.g., favoring reliance on implicit world knowledge to answer), **our evaluation protocol is robust with three safeguards**:
> - **No external knowledge instruction**: As stated in our evaluation instructions (Appendix A.4, \#Line 1067-1068), the evaluator MLLM is explicitly instructed: *DO NOT use any external resources like world knowledge.* This discourages guessing based on prior knowledge.
> - **Refuse to answer (add another "E: None" option)**: As stated in \#Line 158-161, the inclusion of the E: None option is crucial. If the image does not provide the necessary visual evidence to answer the question, the evaluator is instructed to select "E: None", preventing forced-choice guessing.
> - **Majority voting**: As stated in \#Line 320, we employ a 5-round voting mechanism where a response is only considered correct if the ground-truth answer is selected in at least 4 of the 5 rounds. This enhances evaluation stability and reduces the impact of random inference.
>
> Finally, to empirically validate that our questions are visually grounded, we conduct a new ablation study: **evaluating the questions without any image input**. The results below show that the overall accuracy drops to 16.9%, lower than the random guess accuracy of 19.8%. **These results indicate that although the model may exhibit some slight biases (e.g., 28.1% vs. 20.0% in Spatial Interaction), their impact remains minimal.** This result also strongly indicates that without visual context, the model cannot deduce the correct answers and often correctly refuses to answer (by selecting "E: None", which is marked as an incorrect answer since the ground truth is A-D). We have added this discussion to Appendix A.9.2.
>
> |Models|Overall|Spatial Foundation|Spatial Perception|Spatial Reasoning|Spatial Interaction|
> |:---|:---:|:---:|:---:|:---:|:---:|
> |No Image Input|16.9|7.3|19.2|13.5|28.1|
> |Random Choice |19.8|19.7|19.7|19.8|20.0|
> |Qwen-Image    |60.6|69.1|60.7|42.4|79.2|
>
> ---
>
> > **Q4: Analysis of potential evaluation bias related to image aesthetics (style/quality), and whether these factors influence scores independently of the model's actual spatial capabilities.**
>
> A4: Thanks for your insightful suggestion. We agree that it is crucial to ensure our benchmark is not biased by image aesthetics or style, but targeted to spatial intelligence evaluation. This has been added to Appendix A.9.2.
>
> Firstly, we would like to clarify that **our evaluation protocol is inherently designed to be style-agnostic or aesthetic-agnostic.** The task of VLM evaluator is to perform simple Visual Question Answering (VQA) based on factual spatial correctness (e.g., "How are the children arranged around the storyteller?"). **The questions have no aesthetic or stylistic preference.**
>
> Secondly, to empirically validate this, **we conduct a new analysis using HPSv3 (Human Preference Score v3 [4], related to aesthetic and stylistic)**, a SOTA model for predicting human preferences. We compare Flux.1-krea-dev [5], a model known for its aesthetics and realism, with other models of similar SpatialGenEval score. We find that **although Flux.1-krea-dev achieves an exceptionally high HPSv3 score, but does not achieve the top score on SpatialGenEval**. This evidents that our benchmark is not biased towards aesthetic or stylistic preferences, but is instead focused on evaluating spatial capabilities.
>
> |      Models     |   HPSv3 Score   |  SpatialGenEval Score (Ours)|
> | :---: | :---: | :---: |
> | Qwen-Image      |   10.40   |  **60.6**  |
> | GPT-Image-1     |   10.67   |    60.5    |
> | Flux.1-krea-dev | **11.10** |    58.5    |
>
> [4] HPSv3: Towards wide-spectrum human preference score. In ICCV. 2025.
>
> [5] Black Forest Labs. FLUX.1-Krea-dev model offers superior aesthetic control and image quality. https://www.krea.ai/blog/flux-krea-open-source-release.
>
> ---
>
> Please let us know if any concerns remain unaddressed; we are happy to discuss them.

---

### Author Response · Authors · 2025-11-25
**General Response and Summary of Revisions**

Dear Reviewers and ACs,

We are grateful to all reviewers and ACs for their time, effort, and constructive feedback on our submission.

We are pleased that the reviewers found our work to be:
1. `well-motivated`, `clear`, `well-document`, `easy to follow` [$\color{Chocolate}\text{vwE4,WonS,WeDg,awqq}$]
2. `diverse evaluations`, `inclusive to fairly evaluate older T2I models and frontier models`, `holistic evaluation` [$\color{Chocolate}\text{vwE4,WeDg,awqq}$]
3. `quality, soundness, and appropriate complexity`, `high quality` [$\color{Chocolate}\text{WonS,awqq}$]
4. `timely and good job` [$\color{Chocolate}\text{vwE4,WonS,WeDg,awqq}$]

To address the concerns raised, we have revised our paper according to your suggestions and uploaded the revised paper. The major changes are as follows:
1. Addition: Human alignment study between human evaluation and VLM evaluation. `Section 3.2`
2. Addition: The results and discussion of the two most recent closed-source T2I models, i.e., Nano Banana and Seed Dream 4.0. `Section 3.1/3.2, Figure 2(c)(d), Table 2/8/10`
3. Addition: Additional references about spatial understanding, T2I models, and T2I benchmarks. `Section 5`
4. Addition: Analysis of the text encoder capability emerges as a key determinant of spatial intelligence. `Section 3.2`
5. Addition: Fine-tuning on the diffusion-based model (SD-XL). `Abstract, Table 6`
6. Addition: Ablation study of the subset in SpatialT2I and trend on data scaling. `Section 4, Figure 7`
7. Addition: Another solution (prompt rewriting) to improvethe  spatial intelligence of current T2I models. `Section A.6/A.7, Table 11`
8.  Addition: A discussion about the choice of Gemini 2.5 Pro to create T2I prompts. `Section A.9.1`
9.  Addition: A discussion about the potential bias from MLLM predictions. `Section A.9.2`
10. Addition: A discussion about the reliance of MLLM for evaluation. `Section A.9.3`
11. Addition: A discussion about future work to involve a promising object-level extension for further improvement. `Section A.11`
12. Modification: Failure cases analysis of the distribution of error types across scenes and models. `Section 3.2`
13. Modification: A more explicit statement of the construction of SpatialT2I. `Introduction, Section 4`
14. Modification: A clearer statement of how refuse to answer works. `Section 2.1/2.3.2`
15. Modification: A clearer statement of how open-source and closed-source models show similar ranking and number during evaluation. `Section 3.2/A.9.3`

We sincerely appreciate your constructive suggestions, which help strengthen our paper a lot. We are happy to provide additional clarifications. Thank you for your time!

Sincerely yours,

Authors.

---

### Author Response · Authors · 2025-12-02
**Summary of Rebuttal and Highlights of Contributions**

Dear AC, SAC, and PC,

We sincerely appreciate your efforts in making ICLR a success. **During the rebuttal process, our submission had received four positive scores (6,6,8,8) by November 26**. The **3 out of 4 reviewers (WeDg, WonS, awqq) participated in the discussion and raised their scores to acceptance**. Another **reviewer (vwE4) didn't respond but gave an initial high score (8)**. To facilitate your assessment, we provide the brief timeline and summary of rebuttal as follows. Hope this is helpful and can reduce your review burdens.

### **1. Timeline of Rebuttal**
Our paper initially received two positive scores from Reviewer vwE4 (8) and Reviewer awqq (6), and two negative scores from Reviewer  WonS (4) and Reviewer WeDg (2). We **provided our initial point-by-point responses before November 21 and follow-up responses before November 25**. Following our rebuttal,

- Reviewer WonS (`Score: 4 to 6, Soundness: 2 to 3, Contribution: 2 to 3`, on `26 November 2025`) stated that "**satisfied with the human alignment** to confirm the evaluator's performance" and **raised the score**.
- Reviewer WeDg (`Score: 2 to 6, Presentation: 3 to 4`, on `26 November 2025`) praised the new data scaling trends, diffusion experiments, and more discussions, noting "Nice finding...I have **raised my score to honor the thoroughness of the rebuttal**".
- Reviewer awqq (`Score: 6 to 8`, on `22 November 2025`) found human studies and prompt rewriting useful, stating "I will thus **increase my score**".
- Reviewer vwE4 (`Score: 8`) didn't respond, but recognized our work from the start.

### **2. Addressed Concerns from Reviewer vwE4 (Initial Score: 8)**
Reviewer vwE4 had raised four questions. Although they didn't respond, we have addressed their concerns. Specifically,
- Q2: The concern about the human alignment study was also raised by other reviewers and has been successfully addressed, as confirmed by their feedback.
- Q1: A discussion on our choice of Gemini 2.5 Pro has been added to Appendix A.9.1, for its strong creative ability.
- Q3: We have added a clarification in Appendix A.9.2 detailing our method for filtering "prompt-only answerable" QAs, and added a "No image input" experiment to verify its slight biases.
- Q4: A discussion on potential evaluation bias about aesthetics/style has been added to Appendix A.9.2, and the additional HPSv3 experiment verifies our evaluation protocol is not biased towards aesthetic or stylistic preferences.

### **3. Highlights of Contributions**

In summary, we propose SpatialGenEval to systematically evaluate the spatial intelligence of text-to-image models, and explore a spatial-aware dataset (SpatialT2I) as a practical data-centric solution to solve spatial intelligence in T2I models. The highlights of contributions are as follows.

- **Timely and Well-motivated.** Exploring spatial intelligence in T2I models "is a timely and good job" (**Reviewer vwE4**) and "well-motivated" (**Reviewer vwE4, WonS, WeDg, awqq**). Specifically, unlike current T2I benchmarks that often overlook complex spatial failures due to short, information-sparse prompts and simple yes-or-no QAs, SpatialGenEval utilizes long, information-dense prompts and omni-dimensional multi-choice QAs to probe deeper capabilities.
- **Holistic Evaluation Protocol**. The designed evaluation protocol is "holistic evaluation" (**Reviewer WeDg, awqq**). The designed T2I prompts are "inclusive to fairly evaluate older T2I models and frontier models" (**Reviewer vwE4**). Specifically, SpatialGenEval involves 1,230 long, information-dense prompts across 25 real-world scenes. Each prompt covers 10 spatial sub-domains (ranging from object position and layout to occlusion and causality) and is paired with 10 corresponding multi-choice questions (totaling 12,300 QA pairs) to evaluate a model’s understanding beyond what to generate, to where, how, and why.
- **Insightful Analysis.** The evaluation of open- and closed-source T2I models are "diverse evaluation results" (**Reviewer vwE4, WonS, WeDg, awqq**) and "enabling deeper probing of models" (**Reviewer awqq**). Specifically, our extensive evaluation of 23 SOTA models reveals a universal performance bottleneck in spatial reasoning. While models excel at basic object composition, their accuracy significantly falls when faced with tasks requiring higher-order spatial understanding, such as comparison and occlusion.
- **From Evaluation to Enhancement.** Beyond evaluation, our explored spatial-aware dataset (SpatialT2I) is "useful" (**Reviewer WonS**) and designed as a practical data-centric solution to improve existing models. Fine-tuning results yield significant and consistent performance gains across diffusion-, AR-, and unified-based models.

Finally, we believe that **our rebuttal has successfully addressed all concerns raised by the reviewers**. Thank you once again for the time and attention to our work. We hope you find this summary helpful.

Sincerely yours,

Authors.

---

### Meta-Review · Area_Chair_QcFj · 2026-01-06

**Summary:**

This paper introduces Everything in Its Place, a new benchmark designed to evaluate the spatial intelligence of text-to-image models using long, information-dense prompts and structured spatial questions. Reviewers broadly agreed that this is a timely and well-motivated contribution. The benchmark goes beyond existing evaluations that rely on short prompts or coarse judgments, and instead probes where, how, and why objects are placed in generated images. The accompanying SpatialT2I dataset and fine-tuning experiments further show that the benchmark is not only diagnostic but also useful for improving models in practice, which AC deem it as a strong aspect.

**Reviewer Concerns:**

Reviewers did raise concerns. The main one was reliance on VLM-based evaluation, given that current VLMs themselves struggle with spatial reasoning. However, the authors addressed this in the rebuttal by adding a careful human-alignment study and showing strong correlation between human judgments and VLM scores. This substantially reduced concerns about evaluation reliability. Other issues, such as clarity of presentation, potential bias toward aesthetics, and the lack of diffusion-based fine-tuning results, were also addressed with additional experiments, analyses, and clearer explanations, leading multiple reviewers to raise their scores.

While the contribution is primarily a benchmark and data-centric study rather than a new model or theory, reviewers converged on the view that the work fills a real gap and will be useful to the community. The benchmark is thoughtfully constructed, the analysis is thorough, and the rebuttal demonstrated strong engagement with feedback. AC recommend acceptance, most appropriately as a poster.

**Reviewer Scores:**

awqq raised to 8

WonS raised to 6

---

### Decision · Program_Chairs · 2026-01-26

Accept (Poster)